# Mapping the endemicity and seasonality of clinical malaria for intervention targeting in Haiti using routine case data

Ewan Cameron[1,2]*, Alyssa J Young[3,4], Katherine A Twohig[5], Emilie Pothin[3,6], Darlene Bhavnani[3], Amber Dismer[7], Jean Baptiste Merilien[8], Karen Hamre[9], Phoebe Meyer[3], Arnaud Le Menach[3], Justin M Cohen[3], Samson Marseille[8,10], Jean Frantz Lemoine[8], Marc-Aurèle Telfort[8], Michelle A Chang[9], Kimberly Won[9], Alaine Knipes[9], Eric Rogier[9], Punam Amratia[5], Daniel J Weiss[1,2], Peter W Gething[1,2], Katherine E Battle[11]*

[1]Curtin University, Perth, Australia; [2]Telethon Kids Institute, Perth Children's Hospital, Perth, Australia; [3]Clinton Health Access Initiative, Boston, United States; [4]Tulane University School of Public Health and Tropical Medicine, New Orleans, United States; [5]Big Data Institute, Li Ka Shing Centre for Health Information and Discovery, University of Oxford, Oxford, United Kingdom; [6]Swiss Tropical and Public Health Institute, Basel, Switzerland; [7]Division of Global Health Protection, Centers for Disease Control and Prevention, Atlanta, United States; [8]Programme National de Contrôle de la Malaria/MSPP, Port-au-Prince, Haiti; [9]Division of Parasitic Diseases and Malaria, Centers for Disease Control and Prevention, Atlanta, United States; [10]Direction d'Epidémiologie de Laboratoire et de la Recherche, Port-au-Prince, Haiti; [11]Institute for Disease Modelling, Seattle, United States

*For correspondence:
dr.ewan.cameron@gmail.com (EC);
kbattle@idmod.org (KEB)

Competing interests: The authors declare that no competing interests exist.

**Abstract** Towards the goal of malaria elimination on Hispaniola, the National Malaria Control Program of Haiti and its international partner organisations are conducting a campaign of interventions targeted to high-risk communities prioritised through evidence-based planning. Here we present a key piece of this planning: an up-to-date, fine-scale endemicity map and seasonality profile for Haiti informed by monthly case counts from 771 health facilities reporting from across the country throughout the 6-year period from January 2014 to December 2019. To this end, a novel hierarchical Bayesian modelling framework was developed in which a latent, pixel-level incidence surface with spatio-temporal innovations is linked to the observed case data via a flexible catchment sub-model designed to account for the absence of data on case household locations. These maps have focussed the delivery of indoor residual spraying and focal mass drug administration in the Grand'Anse Department in South-Western Haiti.

## Introduction

Malaria transmission in Haiti is endemic and poses a significant public health problem with a total of 8828 cases (presumed and confirmed) reported in 2019 (*World Health Organization, 2019*). However, in relative terms, transmission rates are low: blood stage prevalence of *Plasmodium falciparum* (*Pf*) is in many areas below 1% (*Lucchi et al., 2014*) and the dominant local vector (*Anopheles albimanus*) is inefficient (being primarily zoophilic and exophagic [*Frederick et al., 2016*]). Malaria elimination is a national priority and an ambition of the National Malaria Control Program of Haiti (or PNCM; abbreviated from its official name in French: Programme National de Contrôle de la Malaria). To this end, the PNCM has built a working strategy around improvements to the surveillance and

management system operating nationally through the health facility and community health worker (CHW) network, and the delivery of information and interventions targeted at sub-national administrative regions hosting identified transmission foci (*Boncy et al., 2015*; *Druetz et al., 2018*). To precisely geo-locate transmission foci and develop an evidence-based risk stratification, the PNCM has collaborated with the Malaria Atlas Project (MAP) and partners. In this study, we describe an important component of this collaboration: the construction of a national-level endemicity map and seasonality profile informed by routine case reports from health facilities.

Recent years have seen great progress in the adoption of Bayesian methods for probabilistic map-making (known as model-based geostatistics; *Diggle et al., 1998*) among the infectious disease and global health research community (*Bhatt et al., 2015*; *Osgood-Zimmerman et al., 2018*; *Zouré et al., 2014*; *Karagiannis-Voules et al., 2015*). The standard form of this technique is an extension to the generalised linear model whereby geographic covariates based on high-resolution satellite imaging (e.g., land surface temperature; digital elevation; nighttime lights) are combined additively with a Gaussian process representing spatially correlated residuals. A suitable link function then provides a non-linear transformation to the mean of the presumed sampling distribution for the geo-located, point response data (e.g., prevalence; incidence; presence/absence), often geographically precise to the scale of individual villages and sometimes even households. In the case of malaria, these methods provide benchmark estimates of transmission intensity (*World Health Organization, 2019*; *Weiss et al., 2019*; *Battle et al., 2019*) for much of sub-Saharan Africa where (1) routine case reporting data have historically been highly incomplete and/or subject to problematic sources of bias (*Rowe et al., 2009*; *Alegana et al., 2020*) and (2) the prevalence of malaria is sufficiently high that national-level parasite surveys can readily be powered to resolve spatial variation (*Alegana et al., 2017a*). In low transmission settings such as Haiti, transmission typically becomes increasingly focalised, and the low prevalence of patent parasitaemia forces community parasite surveys towards very intensive (viz. expensive) sampling designs to achieve confident spatial stratifications. Spatio-temporal risk modelling from data deriving from a routine passive surveillance process, such as the reporting of health facility case counts, may thus be a more effective means of describing the heterogeneity of malaria in this setting.

There are many challenges to overcome to achieve accurate, fine-scale disease mapping from health facility case data. Foremost of these is that the case count from a given facility represents the aggregation of observable case incidence over all households over an area of unknown extent: the health facility catchment. Extension of the MGB framework requires the development of a sophisticated sub-model linking the fine-scale disease process with the aggregate data (*Wilson and Wakefield, 2020*; *Taylor et al., 2018*; *Sturrock et al., 2014*), including a representation of health facility choice and attendance (*Duncan et al., 2016*; *Nelli et al., 2020*). Further challenges include a lack of information regarding spatio-temporal variations in treatment seeking propensities across the studied communities (*Alegana et al., 2017b*; *Alegana et al., 2012*; *Battle et al., 2016*; *Karyana et al., 2016*) and in the diagnostic practices operating at each health facility (*Bastiaens et al., 2014*). Validation of model outputs from this class of 'down-scaling' models is also uniquely challenging; the hold-out of aggregate response data is of limited utility for testing fine-scale accuracy, since only ancillary point-level observations can overcome the potential for 'ecological fallacy' (*Wakefield and Smith, 2016*). Complementary to research in this direction is the development of survey methodologies and analysis strategies for alternative diagnostic technologies. For example, serological tools that quantify immune responses to particular malaria antigens can reveal whether or not an individual has ever carried a *Pf* parasite infection (*Corran et al., 2007*; *Helb et al., 2015*), effectively targeting a higher prevalence objective (i.e., lifetime exposure history instead of current infection status) to gain statistical power from lower sampling variance at the expense of temporal precision. Data of this nature have been gathered in Haiti and used in various ways to assist with malaria risk stratification (*Oviedo et al., 2020*).

Here we present the results of a bespoke analysis designed to uncover the characteristic spatial pattern of contemporary malaria endemicity in Haiti and its spatio-temporal seasonality profile using a geostatistical model informed by routine case incidence reports at monthly cadence assembled from across the country over a 6-year period (2014–2019 inclusive). The methodological framework developed for this purpose is described in detail, and model validation against a school-based serological survey is also presented. Finally, we discuss the use of these maps to focus the delivery of

indoor residual spraying (IRS) and mass drug administration in the Grand'Anse department in South-Western Haiti.

## Results

### Fine-scale endemicity surface

Our geostatistical model-based estimate for the contemporary spatial pattern of annual malaria endemicity in Haiti is displayed at 1 × 1 km resolution in *Figure 1*. *Figure 1A* shows our posterior geometric mean estimate of the clinical incidence rate in units of cases per 1000 person-years-observed (PYO), with the fitted data (representative case totals) illustrated by the scaled circles over-laid for those facilities with non-zero case counts. The corresponding clinical incidence surface (i.e., incidence rate multiplied by population) is shown in *Figure 1B*, and a summary of the model-based uncertainties (namely, the pixel-wise standard deviation in our predictions in the logarithm of the clinical incidence rate) is shown in *Figure 1C*. As described in the Materials and methods section below, the representative case totals against which the model was fitted were constructed algorithmically by a procedure designed to (1) clean the data of epidemic fluctuations, (2) impute missing months of data for facilities with reporting gaps, (3) standardise reports towards a diagnostic benchmark of diagnosis by rapid diagnostic test (RDT), and (4) de-trend earlier years of data towards 2019 transmission levels. Details of the spatial and spatio-temporal covariates, treatment seeking surface,

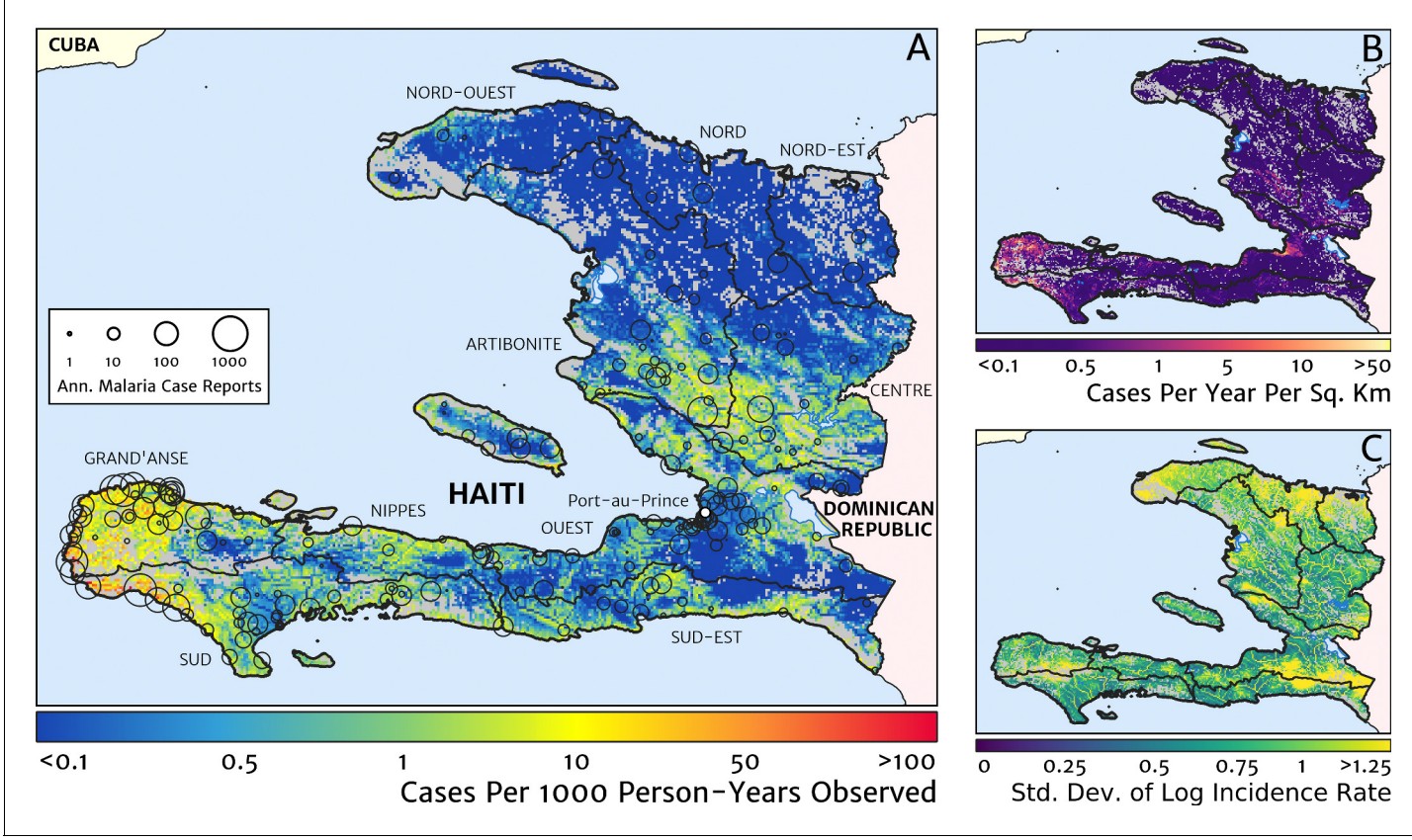

**Figure 1.** The contemporary spatial pattern of malaria endemicity in Haiti (2019) based on reported health facility case counts from 2014 to 2019. (**A**) The (pointwise) posterior (geometric) mean of the clinical incidence rate of malaria in Haiti at 1 × 1 km resolution based on our model fit to 'representative' annual case totals constructed from the health facility dataset. The grey-shaded regions have zero mapped population density, so we do not predict malaria risk in those areas. The boundaries and names (in light capital letters) of the 10 administrative departments of Haiti are marked for reference, as is the location of the capital city, Port-au-Prince. (**B**) The (pointwise) posterior (geometric) mean of the clinical incidence count (total annual cases): this is the product of the risk surface in (**A**) with the population surface. (**C**) A visualisation of the model-based uncertainty in these fine-scale predictions, shown here in terms of the (pointwise) standard deviation in the logarithm of the predicted case incidence rate.

and population map, which are leveraged towards resolution of the endemicity surface below the health facility catchment scale, are also provided in the Materials and methods section.

Two additional visualisations of the inferred clinical incidence distribution in Haiti are provided in *Figures 2* and *3*. In *Figure 2*, we present exceedance and non-exceedance maps at the thresholds of 1 case per 1000 PYO and 1 case per 10,000 PYO, respectively; these illustrations summarise the posterior probability that the incidence rate in each pixel lies above (or, conversely, below) each threshold, and have been identified in previous work on disease mapping as useful summaries for policy-makers (*Giorgi et al., 2018*). In *Figure 3*, we illustrate the aggregate counts of the population-at-risk by department and commune using the same threshold as in our exceedance map; that is, the total number of individuals in each administrative unit estimated to live in areas subject to a case incidence rate above 1 case per 1000 PYO. The first administrative division of Haiti is comprised of 10 departments, and for reference, the names of these are marked (in light capital letters) in *Figure 1A*.

These probabilistic maps of clinical incidence reveal a high degree of heterogeneity in the disease burden due to malaria in Haiti. Large areas of the country – in particular, in the northern departments of Nord-Ouest, Nord, and Nord-Est, and along the Chaîne de la Selle mountain range tracing the border of the Ouest and Sud-Est departments – are essentially malaria free with fewer than one in 10,000 individuals expected to experience clinical malarial each year. Yet there remain a number of high burden communities with clinical incidence rates 500 times greater than this benchmark. These high burden communities are located primarily along the coastline and rivers of the Tiburon peninsula containing the Grand'Anse, Sud, and Nippes departments, with populations-at-risk (defined as those living in an area of malaria incidence greater than one case per 1000 PYO) of 322,693 (95% CI: 280,707–372,057), 322,956 (95% CI: 202,462–392,047), and 108,077 (95% CI: 61,620–147,288), respectively. An additional area of lower but still substantial burden lies within the central river valley joining the Artibonite and Centre departments, with populations-at-risk of 174,766 (95% CI: 97,196–313,169) and 166,938 (95% CI: 95,070–281,816).

The broad credible intervals around the estimation of these populations-at-risk reflect in large part the systematic uncertainties of the de-trending, RDT-standardising, and imputation component of our model, which contribute a substantial variance to inference of the absolute clinical incidence rate, but less so to its relative spatial distribution. Inspection of the uncertainty summary in

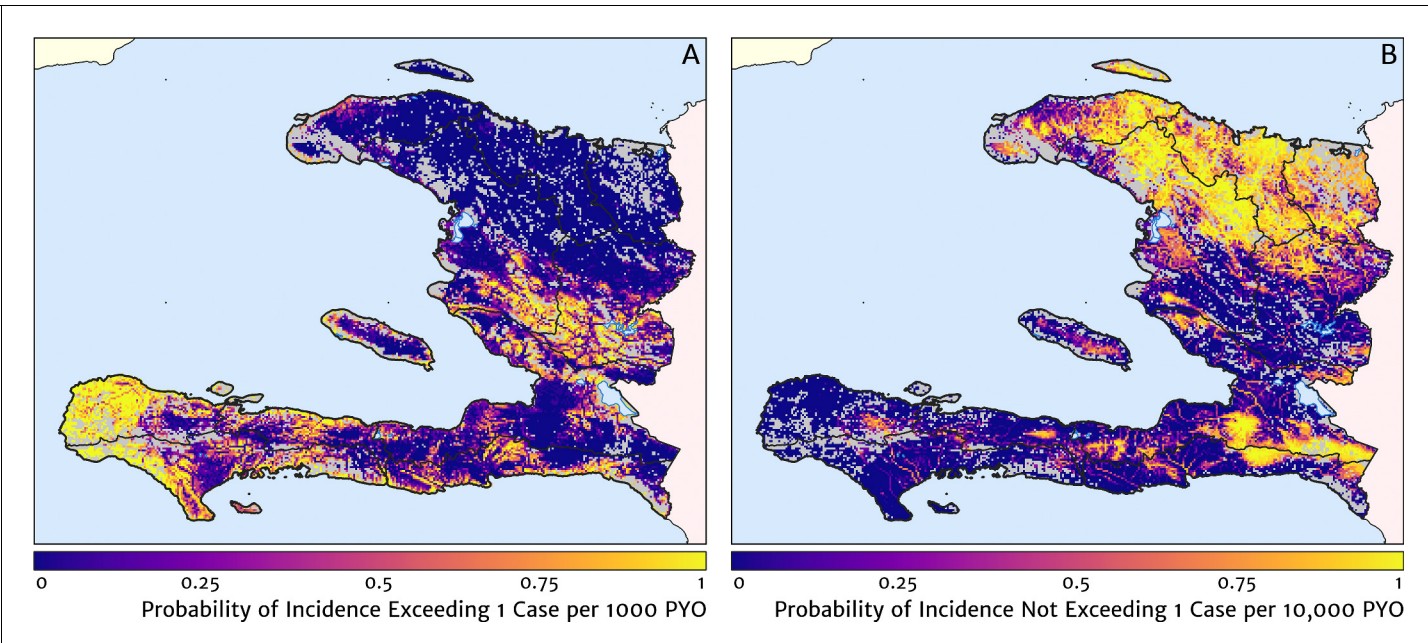

**Figure 2.** Exceedance and non-exceedance maps for clinical malaria incidence in Haiti (2019) at policy-relevant thresholds. (**A**) The posterior probability that the clinical incidence rate exceeds 1 case per 1000 PYO in each pixel under our geostatistical model. (**B**) The posterior probability that the clinical incidence rate does not exceed 1 case per 10,000 PYO in each pixel under our geostatistical model.

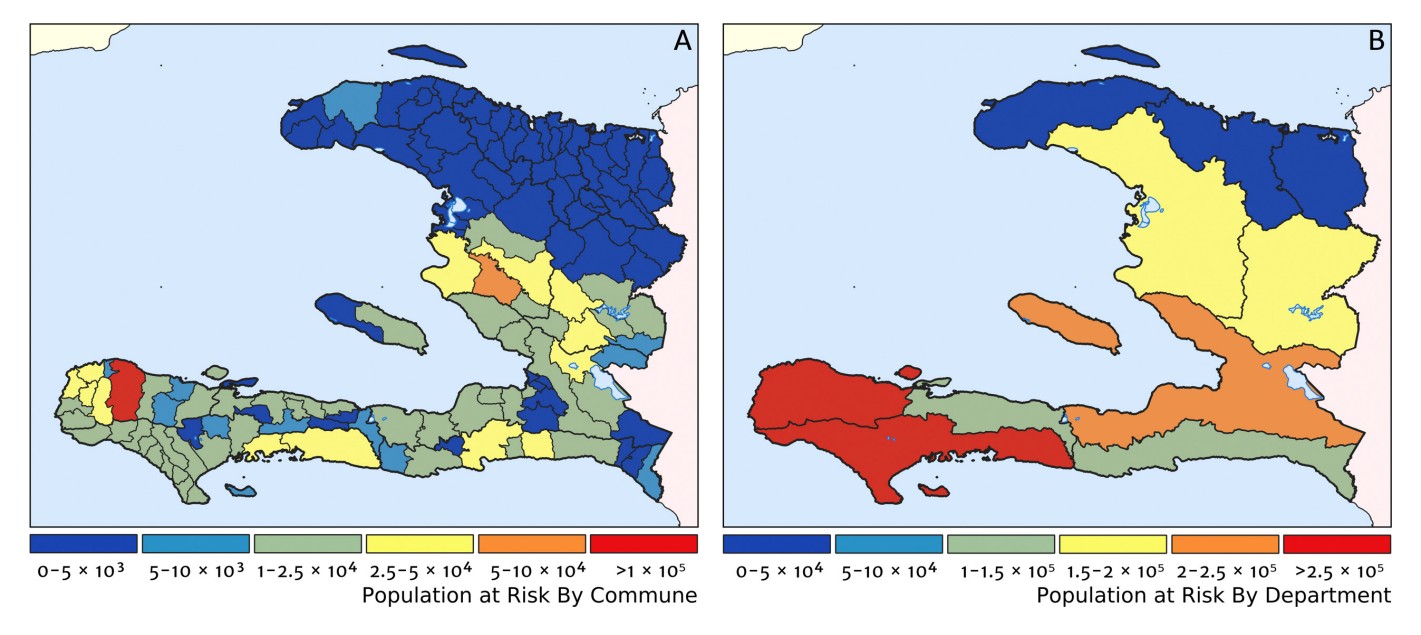

**Figure 3.** Predicted population-at-risk of clinical malaria for Haiti (2019) by commune and department. (**A**) The posterior median estimate of the number of individuals in each commune (the third largest sub-national administrative level in Haiti) living in areas subjected to a clinical incidence rate above 1 case per 1000 PYO. (**B**) The same but aggregated at the level of departments (the largest sub-national administrative level in Haiti).

*Figure 1C* indicates regions of particularly high variance corresponding to (1) the Chaîne de la Selle mountain range and (2) patches along the borders of the Artibonite department and in Nord and Nord-Ouest departments. The explanation for the former is simply population sparsity (i.e., sampling noise) combined with the extreme elevation (i.e., covariate slope uncertainty); however, since this terrain is believed to be an unlikely habitat for the local Anopheles species (principally *A. albimanus*) (*Frederick et al., 2016*), it is probably well-classified as low risk. The explanation for the latter is the ongoing use of microscopic diagnosis at similar frequency to RDT diagnosis at a minority of health facilities in these areas (most used RDT diagnosis near-exclusively in 2019), which leads to a higher contribution here from uncertainty in our standardisation procedure. The accuracy of microscopic diagnosis in Haiti has previously been characterised as inadequate (*Landman et al., 2015*; *Weppelmann et al., 2018*), hence the importance of attempting to adjust statistically for diagnostic type.

The estimation of fine-scale spatial patterns below the extent of health facility catchments is driven within our model by the suite of ancillary environmental covariates. The adopted model structure treats these as linear predictors having slopes that vary spatially with a certain degree of smoothness (as described in detail in the Materials and methods section). The most important covariates under the fitted model for the annual malaria incidence rate are highlighted in *Figure 4*, which shows the dominant positive and negative covariate in each pixel. It is interesting to note that of the four most important covariates over the entire country, three are 'topological' in nature – elevation, accessibility, and road presence/absence – and only one is climatic (potential evapotranspiration). However, it is essential not to interpret these results as indicative of importance in a causal sense; *Figure 4* is presented purely to provide insight into the structure of the fitted regression model. A method for ranking variables in a causal framework has recently been applied to the modelling of malaria case count data from health facilities in Madagascar and the results were shown to be very different to a regression-based variable selection method (*Arambepola et al., 2020*). Note also that the spatially varying slope parameter fitted to each covariate may even change sign in different parts of the country under our modelling framework. For instance, a positive slope is assigned to penalise high elevations in the Chaîne de la Selle mountain range of the Ouest and Sud-Est departments, while in Grand'Anse, a negative slope is assigned to boost the predicted incidence along the (low-lying) coastal fringe.

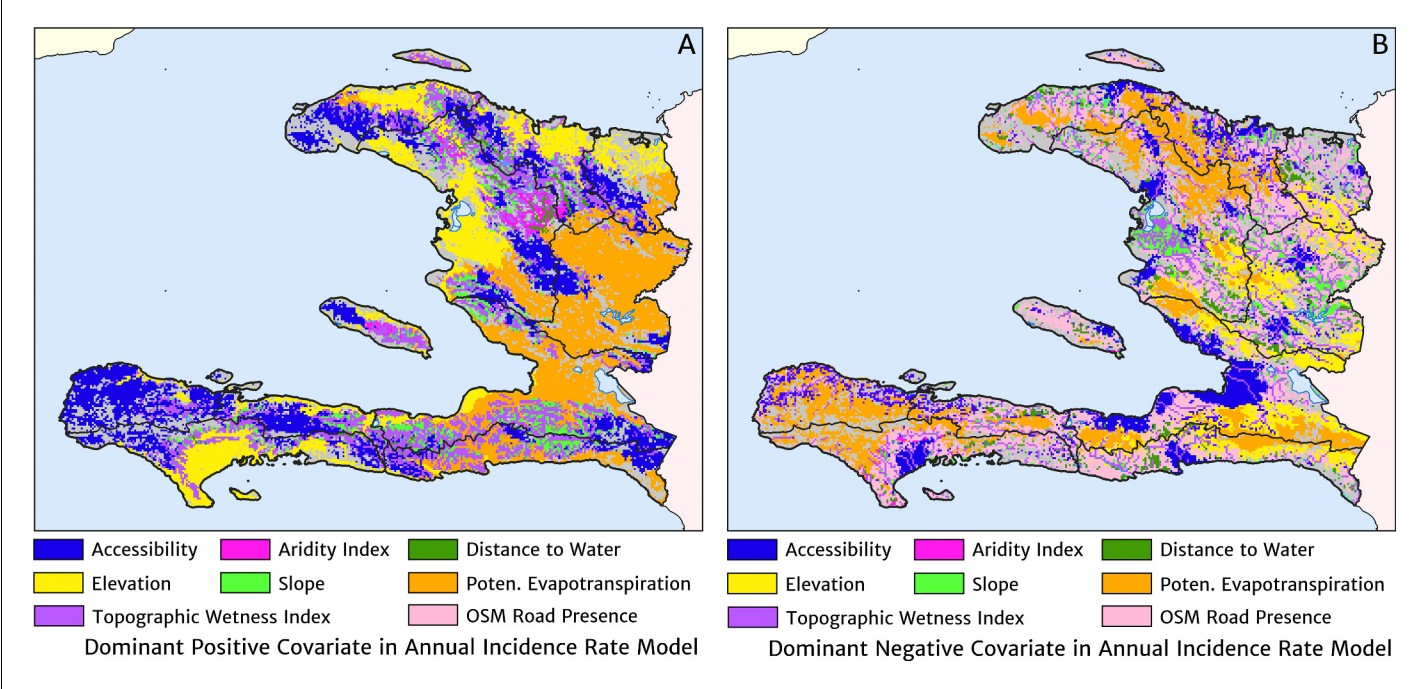

**Figure 4.** The dominant covariates in fine-scale prediction of the case incidence rate for Haiti (2019). The colour of each pixel corresponds to the covariate with (**A**) greatest positive impact (in terms of increasing the local estimate of malaria risk) and (**B**) greatest negative impact (in terms of decreasing the local estimate of malaria risk), upon the predicted incidence rate in accordance with the legend. Of the 12 total spatial covariates offered to the model, only the eight shown here appear among the most dominant in at least one pixel.

## Health facility catchments

The structure of the catchment model used here (see the Materials and methods section for details) allows for patients in any given location to split their attendance between multiple neighbouring facilities according to the relative travel time to each (fixed input) and a relative attractiveness weight (free parameter learnt during fitting). The resulting catchments may thus have overlapping boundaries, which avoids unrealistic structural effects – such as a systematic under-estimation of city health facility patient populations when commuters may otherwise be erroneously assigned to exclusively visit suburban facilities – but can be a challenge for visualisation. In *Figure 5*, we present one type of visualisation of the fitted catchment model: a flow diagram indicating the inferred movement paths connecting the latent (unobserved) case household locations to the reported case counts at health facilities. The accumulated number of cases on each path is represented by a varying line thickness; facilities for which no malaria cases are reported are also indicated without attached flows, for reference. Aside from illustrating the degree of overlap between catchments inherent to our chosen model structure, the visualisations in *Figure 5* highlight the role of the travel-time distances (based on the human movement friction surface of *Weiss et al., 2018*) in shaping these catchments – the connections between inferred case locations and their attended health facilities directly reflect the network of roads linking the settlements of this region.

## Seasonality profile

Our model-based estimates of the fine-scale spatial pattern of month-specific variations in the incidence rate of clinical malaria in Haiti are illustrated in *Figure 6*. For each calendar month, we present the offset (from the annual mean) in the logarithm of the clinical incidence rate surface at $1 \times 1$ km resolution, as fit to the monthly case counts at each health facility in our representative dataset. The dominant signal is a uniphasic seasonal profile evident across most of the country, and most notably the central valley, with cases rising during October–November to a peak in December–January and then declining rapidly from February to a low during April–May. A small number of locations –

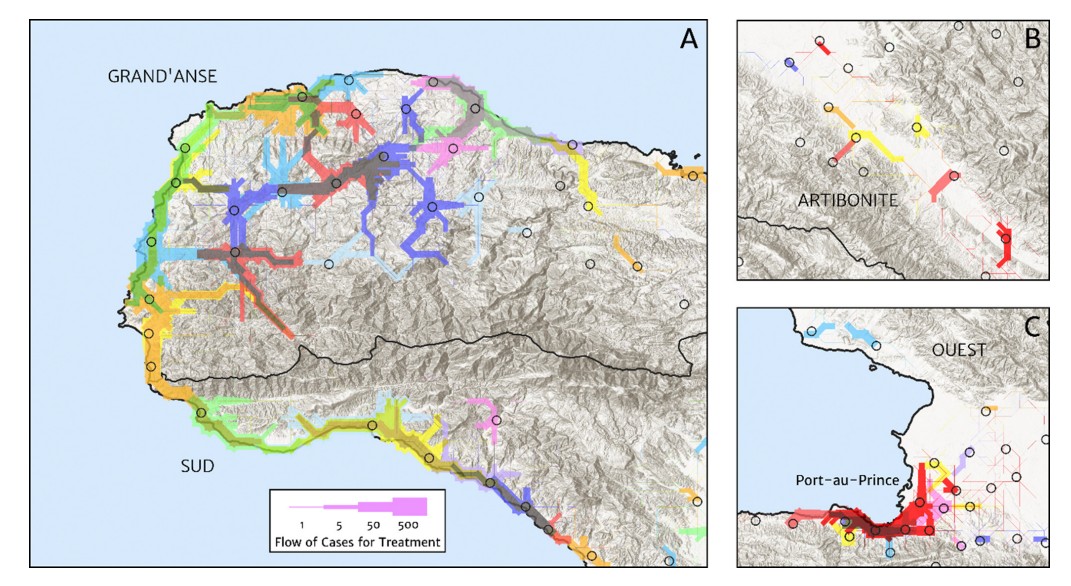

**Figure 5.** Flow paths from predicted malaria case household locations to health facilities based on our catchment model for treatment seeking in Haiti (2019). Each health facility is assigned a random colour and the flow of patients from households to health facilities predicted under our posterior mean catchment model and case incidence surface are illustrated by the colour-matched (semi-transparent) lines of logarithmically proportional thickness, for regions of interest: (A) in Grand'Anse (tip of the Tiburon peninsula); (B) along the Artibonite River in the central valley; and (C) in Port-au-Prince and its surrounding settlements. Note that the flows shown here are modelled at a discretise 1 × 1 km resolution, far coarser than that of the hill-shading relief and coastline shapefiles used in plotting; no journeys by sea are allowed in our least cost path model.

notably one hotspot on the northern coast of Grand'Anse near the town of Jeremie – show a tentative suggestion of a biphasic character with a smaller, second peak in June. However, it is possible that this is an artefact of the recent epidemic outbreak in this area that has not been entirely resolved by our cleaning and imputation procedure for constructing a dataset of representative (endemic character) case counts.

The role of the spatio-temporal covariates that help to shape the estimated seasonality patterns in our geospatial regression model (see the Materials and methods section for details) is explored in *Figure 7*, in which the covariate having the greatest positive influence on the monthly offset in any month is indicated in *Figure 7A*, while the covariate having the greatest negative influence is indicated in *Figure 7B*. In the high incidence areas of the central valley and the Grand'Anse, the most important positive covariate in a predictive sense is the enhanced vegetation index (EVI) lagged by 2 months, while the most important negative covariate is the land surface temperature (LST) lagged by 1 month in the former, and the EVI unlagged in the latter. Again, note that although these covariates may plausibly reflect physical drivers of malaria incidence in Haiti, we caution against this direct interpretation as the fitted model is designed from a predictive framework rather than one of causal inference. Moreover, all of these climatic and vegetation cover covariates are themselves highly colinear, so upon exclusion of one there is typically another able to be identified as providing an explanatory contribution of similar magnitude within the regression model.

## Validation against a serology dataset

The empirical spatial pattern of malaria exposure history amongst children in the Integrated Transmission Assessment Surveys (TAS) for lymphatic filariasis and malaria [*Knipes et al., 2017*] is illustrated in *Figure 8A*. The TAS program used serological markers of long-term malaria exposure – apical membrane antigen (AMA) and merozoite surface protein (MSP) antigenic responses – to characterise malaria endemicity in school-aged children. The symbols plotting the TAS results in *Figure 8* are colour-coded with respect to a (non-geospatial) Bayesian estimate of the median underlying sero-prevalence (positivity by either antigenic response, or both) at the location of each school

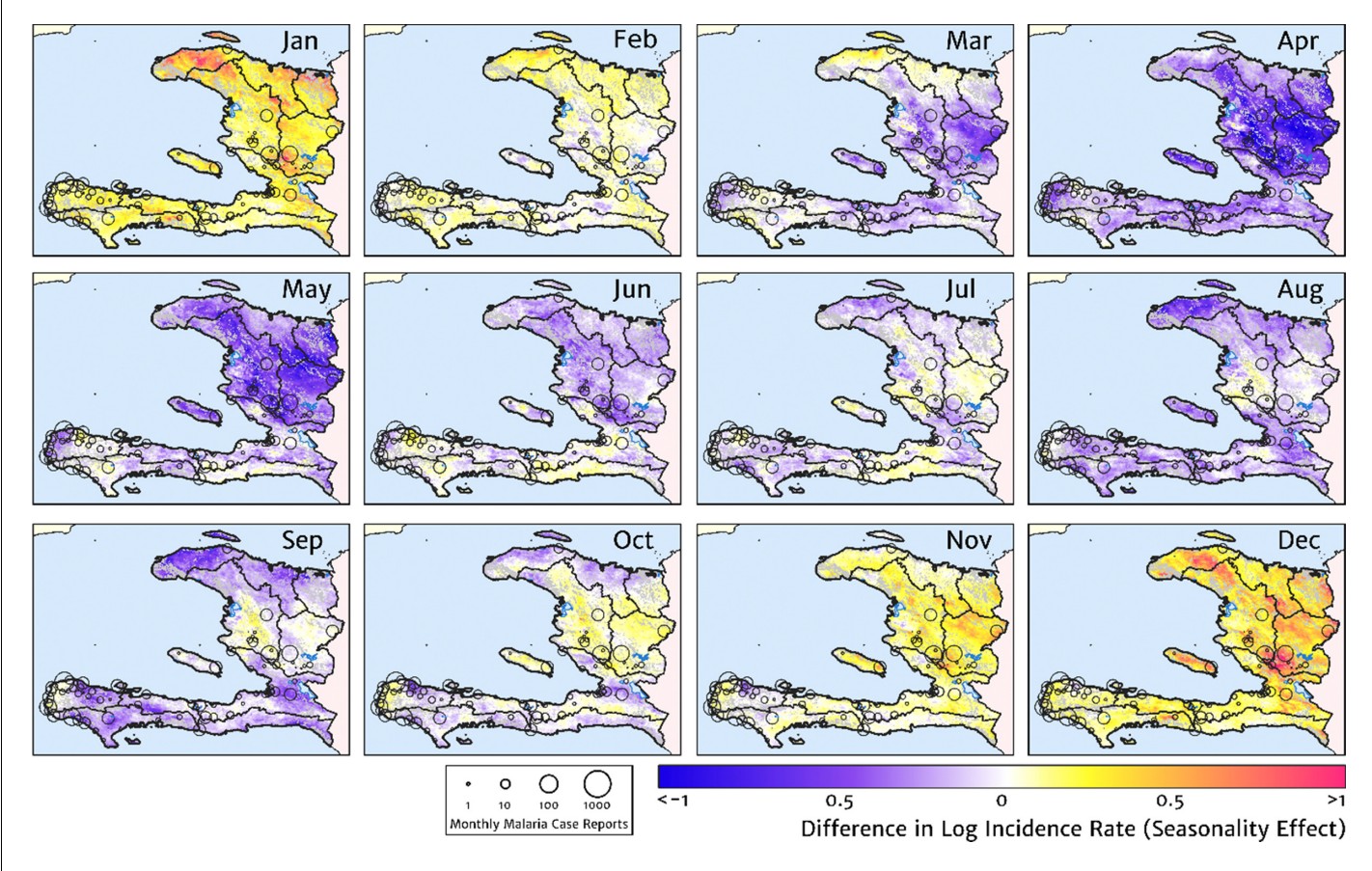

**Figure 6.** The typical fine-scale spatial pattern of month-specific variations in the incidence of clinical malaria in Haiti based on reported health facility case counts from 2014 to 2019. The (pointwise) posterior mean of the seasonal effect on the logarithm of the predicted case incidence rate is illustrated for each calendar month based on the third stage of our inference procedure: the spatio-temporal geostatistical model with fixed catchment sub-model fitted to the representative monthly case counts constructed at health facility level.

surveyed. These estimates may be compared visually in a geographic context against our predicted clinical incidence rate surface in *Figure 1A*. In Panel B, we present an alternative graphical comparison via a scatter plot (magenta 'crosses') with the uncertainties in each metric shown as error bars. The median-to-median correlation between these two metrics of transmission intensity has a Pearson coefficient of 0.426 (95% CI: 0.353–0.499), reflecting a positive but noisy relationship. Of course, since the TAS sample size per school is generally small (<30), the credible intervals around these point estimates of sero-prevalence are correspondingly broad. When considered in addition to the uncertainties surrounding the fine-scale predictions from our health facility-based model, it is likely that much of the width in this scatter plot derives from random (sampling) noise. Aggregating the TAS sites in bins of similar predicted clinical incidence rate reveals a much tighter relationship, for which a simple linear regression of the logit of sero-prevalence against the (natural) logarithm of incidence yields a slope of 0.34 (i.e., $\operatorname{logit} p_{\text{AMA or MSP}} \propto 0.34 \times \log I$).

Where visual comparison of *Figure 8A* to *Figure 1A* indicates the most interesting discrepancy is with regard to the presence of some moderate (and in one case, high) sero-prevalence schools in the Nord department for which our predicted clinical incidence rate from the health facility dataset is everywhere rather low. We suspect that this is a reflection of a strong decline in transmission intensity in this region over the period 2014–2016, seen in the rapid decline in cases reported and hence the strong de-trending in our model towards the 2019 case counts – although it is not possible to unambiguously distinguish changes to the health reporting system from genuine transmission intensity trends from the available data. An important point raised by this comparison is that the incidence surfaces presented here should be understood as reflecting the current state of transmission

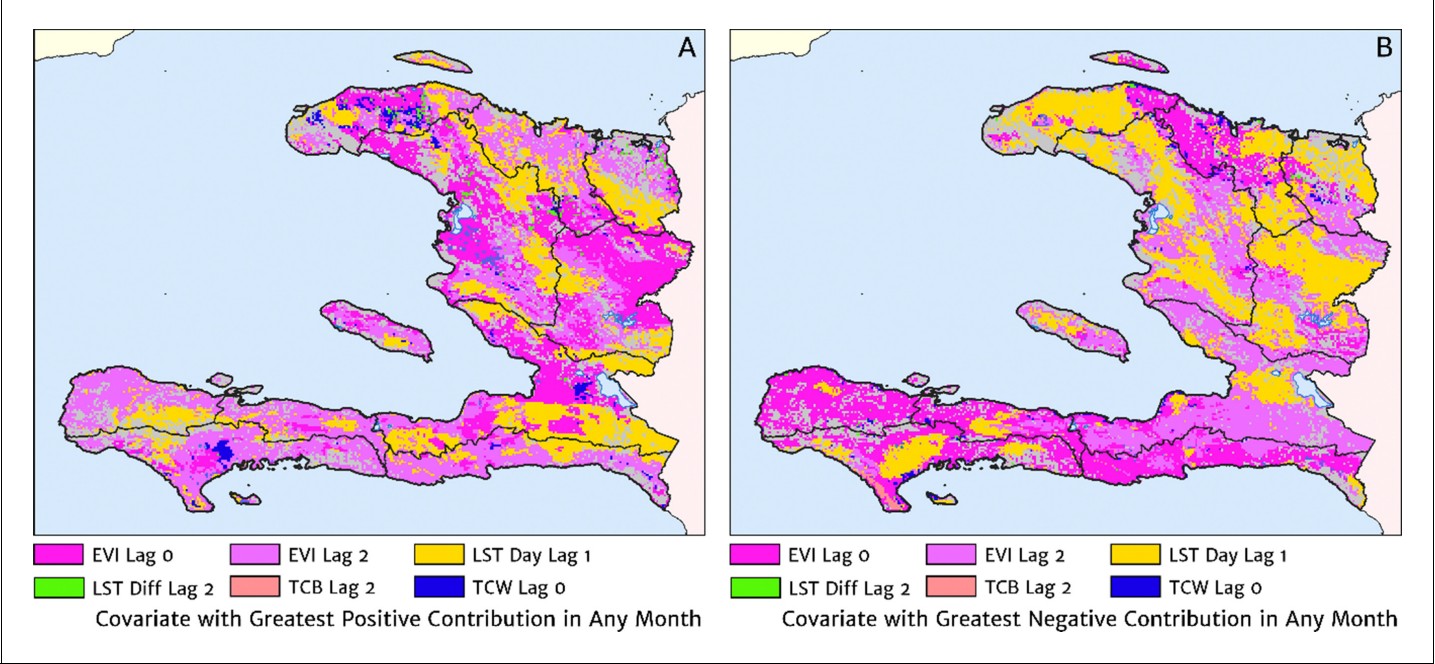

**Figure 7.** The dominant covariates in fine-scale prediction of month-specific variations in the incidence of clinical malaria in Haiti (2014–2019). In each pixel, the colour key indicates the covariate having the greatest (A) positive or (B) negative influence on the monthly incidence offset in any month. The lags denoted here are in units of months prior.

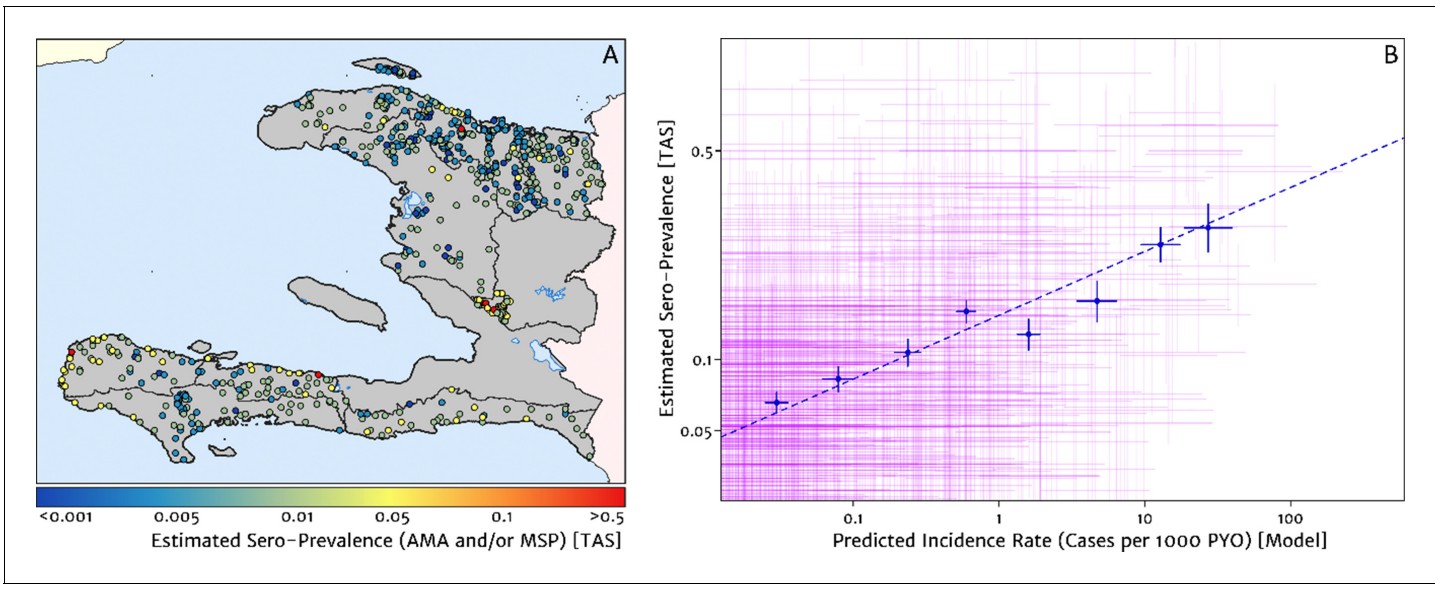

**Figure 8.** Model validation against the estimated proportion of school children testing positive to serological markers of past malaria exposure in the TAS dataset (2014–2016). (A) The spatial location of each school sampled in the TAS study is illustrated here with the colour of the plotting symbol (filled circle), indicating the estimated sero-prevalence at that site. In this case, sero-positivity is defined as being classified positive for either the MSP antigenic response, the AMA antigenic response, or both. (B) Comparison of the estimated sero-prevalence (using a simple Bayesian beta-binomial model) from the TAS schools data against the predicted case incidence rate from our full geospatial model fit to the representative health facility-level data. The 95% credible interval in each metric for each school location is illustrated by the purple lines. The median estimated sero-prevalence for sites grouped in a series of bins by predicted case incidence is overlaid in blue, along with the associated line of best fit.

subject to the recent history of anti-malarial interventions, rather than as a reflection of the inherent environmental receptivity under a zero intervention scenario.

## Comparison against models with naïve imputation and naïve catchment structure

A minimal alternative approach to risk mapping from routine case data that has been explored in the past was to impute missing case reports using the empirical mean over non-missing months (on a per-facility basis) and to attribute cases from each facility to the households for which that facility is the nearest treatment option, ignoring differences in the diagnostic method (microscopy vs RDT). Following this procedure for the case data from 2019, we recover the risk map shown in *Figure 9A*. The effective 'resolution' of this map is to the size of each naïve catchment area, which tends to be smaller in towns and cities and larger in remote, rural areas. Compared with our preferred model-based risk map (*Figure 1*), the result here would suggest that transmission in the Grand'Anse is dominated by hotspots sharply concentrated on the townships of Abricots, Bonbon, Anse-d'Hainault, and Les Irois, with transmission intensity above 100 cases per 1000 PYO in each, rather than being spread more evenly throughout the coastal settlements and rural communities of this peninsula. The correlation coefficient of this risk map against the TAS sero-prevalence data is just 0.301 (95% CI: 0.215–0.369), compared against 0.426 (95% CI: 0.353–0.499) for our preferred model. A second version of the naïve catchment risk map is shown in *Figure 9B*, this time after using our Step one and Step two models (see Materials and methods) for de-trending, imputing, and diagnostic correcting the raw case data. Overall, this adjustment improves the correlation against the TAS sero-prevalences to 0.331 (95% CI: 0.257–0.398).

## Discussion

The fine-scale mapping of malaria incidence and its seasonality profile in Haiti achieved through our fitting of a Bayesian geospatial regression framework with catchment sub-model to the 2014–2019 health facility case reports brings a greatly refined understanding of the elimination challenge on this side of Hispaniola. We see that the communities suffering from the highest annual average rates of clinical malaria (above 50 cases per 1000 PYO) in 2019 are those along the coastline and valleys of

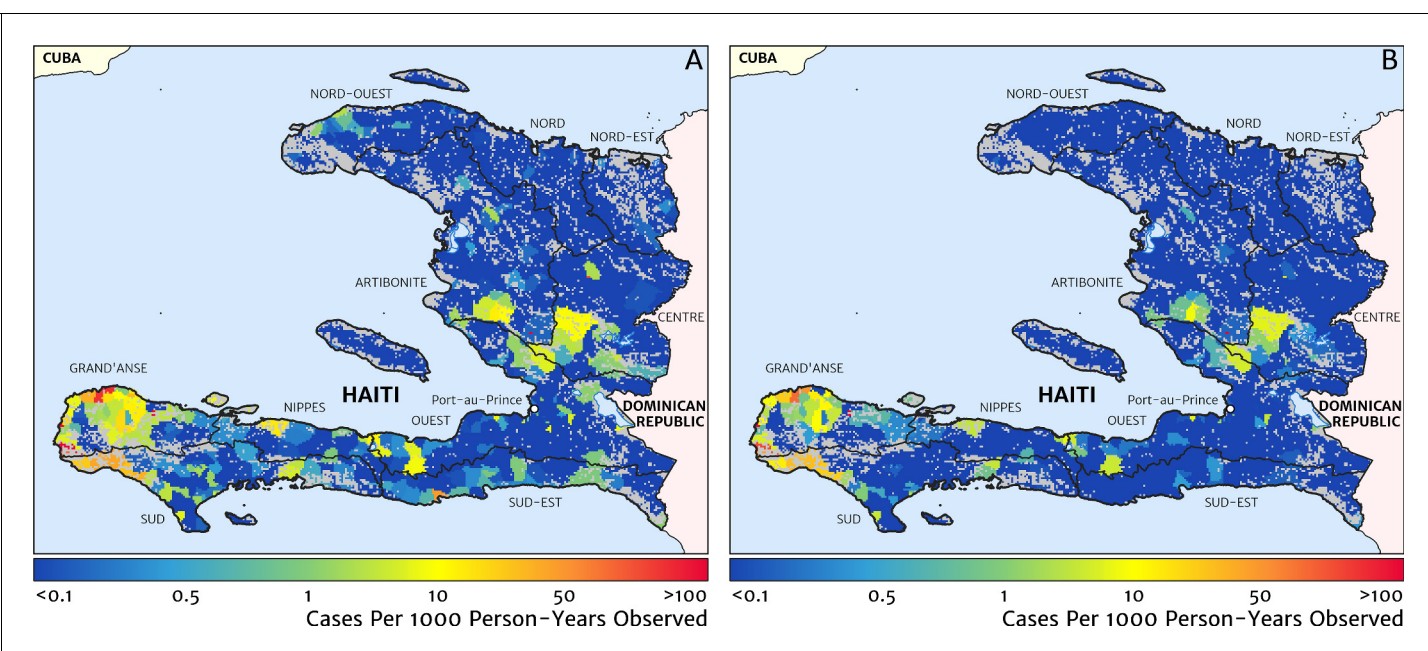

**Figure 9.** Risk stratification maps for 2019 produced under a naïve catchment model in which patients attend only their nearest facility. (**A**) The raw case data is used with a crude imputation by way of per-facility empirical means excluding missing months; (**B**) the case data has now been imputed, de-trended, and microscopy-to-RDT converted.

the Grand'Anse and Sud departments. Additional pockets of low-to-moderate endemicity (1–10 cases per 1000 PYO) are located in the central valley spanning the Artibonite and Centre departments, in some coastal communities of the Nippes, Sud-Est, and Nord-Est departments, and surrounding Port-au-Prince in the Ouest department; the latter accounting for a substantial proportion of the total cases each year owing to the size of the population in this area. The Nord and Nord-Est departments have lower incidence rates (below 1 case per 1000 PYO), and some areas can yet be confidently predicted as extremely low (below 1 case per 10,000 PYO). Against these broad variations between departments, there exists substantial heterogeneity in the clinical incidence rate of malaria at rather small scales within departments, which in a predictive sense can be explained within our modelling approach by differences in accessibility, elevation, road presence/absence, and potential evapotranspiration. The clinical incidence of malaria in Haiti is also highly seasonal with a strong uniphasic seasonality pattern at maximum during December–January and minimum during April–May.

These results have already proven useful for planning a number of the public health interventions that will be required to achieve malaria elimination in Haiti. These maps have been used to derive epidemiologically relevant operational units for targeting packages of interventions in five priority communes in Grand'Anse. Operational units were ranked by the strength of transmission (quantified after further post-processing in terms of the reproduction number under control, $R_c$) to help determine those that would receive targeted mass drug administration (tMDA) and with IRS in 2018, and are again being used in planning this year (2020). Serological data were subsequently used to refine this ordering, and an eventual re-definition of operational unit boundaries was made to follow natural logistical divisions such as rivers and major roads. For planning purposes such as these, it is clear that these fine-scale probabilistic maps offer a more nuanced stratification than the categorical risk maps at the commune level produced, e.g., for the World Malaria Report (*World Health Organization, 2019*), by direct summary of the available case counts divided by areal population totals – and one that is far superior to the dichotomous risk maps based solely on elevation (at a threshold of 500 m) that have, anecdotally, been used in past decision making.

In this context, it is important to again emphasise certain caveats of our analysis, which point towards topics for future research and data gathering. Of particular concern is the lack of information regarding potential spatial variations in treatment seeking behaviour across the country. A recent study of community attitudes towards malaria treatment (*Druetz et al., 2018*) confirmed that some Haitians will seek care for febrile illness outside the national health care system, such as at a traditional healer or at a private health care provider not reporting to the national network. At present, we have attempted only to adjust for the possible effect of travel-time distance on the absolute treatment seeking propensity, using a model calibrated to an African setting (*Alegana et al., 2012*); clearly, this deserves refinement if local data can be gathered through a dedicated survey questionnaire.

It is known that seasonal migrations of agricultural workers or other large itinerant groups have a potential to introduce spurious effects into modelled case incidence rates unless explicitly accounted for via a dynamic population denominator (*Zu Erbach-Schoenberg et al., 2016*). As high-fidelity human movement data is not currently available for Haiti, we cannot yet model this aspect directly and can only hope that a substantial proportion of any such unmodelled variation is absorbed implicitly within the random effects terms of our statistical model. Interestingly, an earlier study in which a regression model was built to predict short-term human movement from internal migration data (*Sorichetta et al., 2016*) has indicated that in relative terms the Ouest department containing Port-au-Prince is more strongly connected to all other departments than any of those departments are connected with each other independently, although the magnitude of this connectivity in absolute terms remains unknown. It is worth emphasising here that our model aims only to map where people at risk of malaria illness reside, which may not necessarily be the same as where they contracted their infection. The higher risk communities identified in our modelling are primarily in remote and rural areas, in which people are unlikely to regularly commute long distances from their place of residence for work or leisure. However, in terms of absolute case numbers it was shown in *Figure 1B* that there are a substantial number of people in the vicinity of Port-au-Prince presenting to health facilities with clinical malaria. Whether these infections were contracted locally or elsewhere – and what role seasonal migration and/or travel plays in sustaining transmission wherever it occurs – is not informed by the present dataset.

A final caveat on our analysis concerns the limitations of the catchment sub-model. Our introduction of a gravity-style representation with overlapping catchments based on travel-time distances constitutes a substantial effort towards constructing a realistic representation of patient behaviours, especially in comparison with the vanilla alternatives of Euclidean ('as-the-crow-flies') distances and/ or hard (tessellation-style) boundaries – yet there is no doubt that our model is still a profound simplification. It remains for future research to establish how much of a limitation this is in terms of our ability to accurately downscale health facility data to pixel level – though at least our comparison against the TAS serological data suggests we are on the right track – and whether there are any simple improvements to the model structure that should be made (such as a refinement of the coefficient of preference decay on travel-time distance, currently fixed at $-2$; i.e., inverse-square decay). Already we have begun work (van den Hoogen et al., in prep.) to explore risk mapping under more complex catchment sub-models in a focus region of the Artibonite department where partial case tracing of febrile patients (from health facility to patient household location) has been performed. The catchments we have begun to reconstruct in the Artibonite case tracing study do confirm a general dependence on travel time, but they also reveal instances in which clusters of patients travel far beyond their nearest facility to seek care. We do not have data on specific factors, which might help to explain this behaviour, though anecdotal examples that appear in the literature suggest possible explanations, e.g., lower income patients may be avoiding a facility that illegally charges for antimalarial medication (*Druetz et al., 2018*).

Another direction we are exploring to further refine our risk maps is the inclusion of information from alternative malaria metrics such as the sero-positvity rate, used here only for model validation. Important to note is that, although our current validation model treats the underlying sero-prevalence at each site as an independent random variable, one can readily apply the same principles of model-based geostatistics to refine sero-prevalence estimates via spatial covariates and spatially correlated noise models (*Ashton et al., 2015*). While we have not taken this step here to avoid any artificial shrinkage of our validation set towards the health facility dataset through a common model structure with the same covariates, it is easily done. More challenging is to develop appropriate methods for the simultaneous modelling of multiple data types. Indeed this is an active topic of research within geospatial statistics – both in regard to linking point data with areal data (*Richardson and Best, 2003*; *Moraga et al., 2017*) and in regard to sharing information between multiple disease metrics (for the same disease or even different diseases [*Held et al., 2005*]) – and is a direction we are pursuing for further refinement of our incidence maps in the Grand'Anse department (Amratia et al., in prep.).

In conclusion, the analyses and results of this paper demonstrate that point of care case counts can be used to generate programmatically useful maps of clinical incidence rates providing fine-scale risk stratifications. A spatio-temporal seasonality profile can also be determined when data are available at monthly intervals. This information can be used to refine the spatial and/or temporal targeting of high-burden areas for anti-malarial interventions such as tMDA, IRS, and long-lasting insecticidal net delivery. These outputs are readily updatable as additional facility data are made available and will be valuable in defining residual transmission foci as the final stages of elimination near.

## Materials and methods

### Response data and covariates

Our primary dataset consisted of monthly counts of confirmed malaria cases – i.e., patients seeking care for febrile illness with patent parasitaemia detected via RDT or microscopy – for each of 771 geo-located health facilities reporting at least once in 2019. These 771 facilities are a sub-set of the 1191 facilities in a master reporting file compiled by the PNCM of Haiti with assistance from the Clinton Health Access Initiative (CHAI); those that did not report on malaria cases or testing at least once in 2019 were assumed here either to have closed or to no longer offer malaria test and treat services to febrile outpatients. Since 2016, CHWs attached to certain health facilities have been proactively seeking cases in the local community, and we add these cases to the reported totals of ordinary patient visits for those facilities. The reporting period covered by this dataset begins with January 2014 and finishes with December 2019 and the overall completeness of reporting among

the sub-set of 771 facilities is 77.5%, with 537 facilities reporting in at least 61 of the 72 months (69.6%). It is believed that all health facilities operating in Haiti through 2019 are included in the master reporting file, although it is not certain that all facilities excluded from the sub-set studied in the present analysis represent genuine closures as opposed to circumstances of sudden, sustained reporting failure, or conversely that all included facilities were indeed open through all of 2019. A number of the health facilities in our dataset lie within the same city or village, being separated by a distance comparable to, or less than, the target resolution (1 × 1 km) of our mapping. After imputing missing monthly case reports for all 771 facilities reporting in 2019 (as described in stage one of our inference procedure below), we reduce the subsequent model complexity by aggregating nearly co-located facilities using a hierarchical clustering algorithm. In this way, a total of 450 'aggregate pseudo-facilities' are formed which we will simply continue to refer to as 'health facilities'.

A suite of high-resolution satellite imaging products were introduced as covariates within our statistical modelling. Namely, accessibility to cities (*Weiss et al., 2018*), aridity index (*Trabucco and Zomer, 2009*), distance to water (bespoke), elevation (*Farr et al., 2007*), EVI (*Huete et al., 1999*), land cover classification (forest, grass savannah, urban/barren, and woody savannah; *Friedl et al., 2010*), LST(day and day–night difference; *Wan et al., 2002*), open street map (2016 road presence/absence; *Haklay and Weber, 2008*), potential evapotranspiration (*Trabucco and Zomer, 2009*), slope (*Farr et al., 2007*), tasselled-cap brightness (TCB; *Kauth and Thomas, 1976*), tasselled-cap wetness (TCW; *Kauth and Thomas, 1976*), and topographic wetness index (*Farr et al., 2007*). All products were downloaded from their respective online repositories, gap-filled (where necessary), and registered to a common grid. The EVI, LST, TCB, and TCW products were summarised to 2014–2019 annual averages and average monthly offsets, while the remainder were used as static covariates. The High Resolution Settlement Layer from the Connectivity Lab at Facebook (URL: https://www.ciesin.columbia.edu/data/hrsl/) provides the population denominator for our model, and the *Weiss et al., 2018* friction surface is used to build travel-time maps from each 1 × 1 km pixel to each health facility with assistance from the *malariaAtlas* R package (*Pfeffer et al., 2018*). Serological prevalence observations (AMA and MSP antigens) for 24,514 children aged 6 and 7 years old in 820 schools from the Integrated TAD datasets (*Knipes et al., 2017*) sampled from across Haiti between November 2014 and August 2016 were used for validation of the spatial trends revealed in the annual incidence outputs.

## Stepwise modelling approach designed for robustness against unmodelled sources of noise

Case counts of clinical malaria from health facilities in low-resource settings have traditionally been considered an unreliable and challenging source of data with which to map risk and/or evaluate the efficacy of interventions, owing to spatial and temporal variabilities in reporting completeness and accuracy, testing rates and methods, access to care, and treatment seeking behaviours (*Alegana et al., 2020*; *Afrane et al., 2013*; *Oduro et al., 2016*; *Ohiri et al., 2016*). In Haiti in particular, the specificity of local microscopy-based diagnosis has been shown to be sub-optimal (*Landman et al., 2015*; *Weppelmann et al., 2018*) and increasing the proportion of diagnoses made by RDT has been a key pillar of recent reforms to case management (*Boncy et al., 2015*) – the impact of which is clearly seen in the reported health facility case counts (*Weppelmann et al., 2018*). Fortunately, in this study, we have access to data on the relative rates of RDT and microscopy testing by health facility and month, allowing for explicit modelling of this previously identified systematic effect. Spatial variation in access to care is another systematic effect that we attempt to model given our access to a high-resolution travel-time covariate (*Weiss et al., 2018*). However, we must acknowledge that there are likely many other important confounding factors about which we have very little supporting data. Likewise, the spatio-temporal dynamics of epidemic fluctuations in malaria incidence are challenging to separate from the signal of endemic transmission intensity via a generative (forward-modelling) framework. The stepwise inference framework described below is designed to limit the impact of such factors on our model-based estimates while negotiating a pragmatic trade-off between the theoretical advantages of building an explicit representation of each conceivable error term and the computational advantages of model parsimony.

Our inference of the fine-scale case incidence rate and seasonality profile of clinical malaria in Haiti is thus modularised in three distinct stages. In the first stage, a pair of statistical models is used to impute missing data and de-trend the reported monthly case counts from 2014 to 2019 at each

facility towards an RDT-standardised 2019 level. These are then folded (over years) and median filtered by month to produce a year of 'representative' case data designed to reflect endemic transmission intensity. Model-based uncertainties from this procedure are propagated through to the subsequent stages of our analysis by sampling multiple versions of this representative dataset from its modular Bayesian posterior. In the second stage, a fine-scale, spatial-only geostatistical model with a flexible catchment sub-model is fit to (each modular posterior draw of) the representative dataset to estimate the annual average incidence rate at pixel-level and the attractiveness of each health facility. Fine-scale mapping in this step is assisted by our suite of high-resolution covariates and an over-dispersed sampling distribution is adopted to represent additional variation in the reported counts beyond that accommodated naturally by our core model. Again the statistical uncertainties are propagated forward via modular posterior sampling. In the third and final stage, a fine-scale spatio-temporal geostatistical model is fit (conditional on the previously fitted catchment sub-model) to explain the residual seasonal variation about (each modular posterior sample of) this baseline risk surface in the (corresponding sample of) representative monthly case data.

Although the primary motivation for introducing these 'cuts' (*Plummer, 2015*) in our inferential approach is, as noted above, to focus on endemic transmission, promote model parsimony, and improve computational feasibility in model fitting, it is worth noting that such contained modularisation can also guard against the magnification of systematic errors between components due to a misspecification in one of them (*Jacob, 2017*). The following sections give further details on each of the three stages.

## Constructing a year of representative case data

The first step in this stage of analysis was to impute values for the fraction of tests performed by microscopy (as opposed to RDT) in those health facilities missing these data in certain months. To this end, we introduce a non-spatially structured model in which the expected proportion of tests conducted by microscopy in each month for each facility is predicted as the inverse logit transformation of a three part temporal spline (covering January 2014 to December 2019) plus intercept. The spline coefficients are assigned a Bayesian shrinkage structure in which the mean of each and the between-facility variation are learned jointly across facilities. The precise structure of this model is described in standard hierarchical Bayesian notation in the box for Model 1 below.

$$N_{\text{mic,jt : where mic and RDT case totals both non-missing}} \sim \text{Binom}\left(p_{\text{mic,jt}}, N_{\text{tested,jt}}\right)$$

$$\text{logit } p_{\text{mic,jt}} = a_j \times \beta_{\text{spline(1)}}(t) + b_j \times \beta_{\text{spline(2)}}(t) + c_j \times \beta_{\text{spline(3)}}(t) + d_j$$

$$a_j \sim \text{Normal}\left(a_{\text{mean}}, \sigma^2_{\text{shrinkage}}\right), b_j \sim \text{Normal}\left(b_{\text{mean}}, \sigma^2_{\text{shrinkage}}\right)$$

$$c_j \sim \text{Normal}\left(c_{\text{mean}}, \sigma^2_{\text{shrinkage}}\right), \log \sigma_{\text{shrinkage}} \sim \text{Normal}(-1, 1^2),$$

$$a_{\text{mean}}, b_{\text{mean}}, c_{\text{mean}}, d_j \sim \text{Improper Uniform}$$

### Model 1

Facility-level model with non-spatially-structured Bayesian shrinkage for the estimation of the month and facility-specific propensity to conduct malaria diagnosis by microscopy rather than RDT.

Our de-trending model then takes the form of a point-indexed geostatistical regression on the case counts, $\text{cases}_{jt}$, at each facility in each month (where available), computed with respect to a latent incidence surface using the associated populations under a naïve catchment sub-model as a base rate factor. For the latter, we propose that the population in a given pixel will split its attendance between neighbouring health facilities in inverse proportion to the square of travel-time distance from pixel to facility. In mathematical notation, our naïve catchment matrix, $C^*_{i \to j}$, which gives the proportion of residents in pixel $i$ who attend health facility $j$ is constructed as $C^*_{i \to j} \propto \frac{1}{T^2_{i \to j}}$ using travel-time distances, $T_{i \to j}$, computed from the Weiss et al. friction surface (*Weiss et al., 2018*). We distinguish this formulation (the naïve sub-model) from the more flexible version introduced in the subsequent analysis stages in which an 'attractiveness' weight, $W_j$, is learnt for each facility during fitting. This weight represents the impact of unknown factors that might influence attendance preference, such as differences in the availability of staff, the cost of treatment, and perceptions about the

quality of care offered. Multiplication of the naïve catchment sub-model against the high-resolution population map for Haiti (while assuming, for now, universal access to treatment) gives crude population denominators for each facility, adequate for this temporally focussed inference step.

The statistical structure of our de-trending model comprised a five-part temporal spline across the 72 months of data with spatially varying coefficients and a spatially varying intercept, as well as a (cyclical) annual seasonality term, as described using hierarchical Bayesian notation in the box for Model 2 below. The mean (log) incidence surface is composed of a spatial-only Gaussian process term and a separable (Kronecker product) spatio-temporal Gaussian process with circularity (over the calendar months) in the temporal dimension (an exponential kernel on the circle). Model fitting was performed in the Template Model Builder (*TMB*) and Integrated Nested Laplace Approximation (*INLA*) packages (*Kristensen, 2015*; *Lindgren and Rue, 2015*) for R using a Laplace approximation over the random field components and over-dispersion terms, and with posterior approximation over the remaining hyper-parameters represented by a Multivariate Normal matched to the curvature at the empirical Bayes estimate. The suitability of this higher level approximation was confirmed by comparing the (Laplace approximation based) marginal likelihoods at a series of draws from the Multivariate Normal against their densities under this proposal distribution. As this is an expensive operation, we did not calculate and use these factors for importance weighting of our full set of approximate posterior samples, relying instead on the nested Normal formulation.

Finally, for each posterior draw, we impute the missing case reports with predicted case numbers and divide from the completed case–month matrix the exponentiated $f_{(k)}(\mathrm{loc}_j) \times \beta^{(k)}_{\mathrm{spline},t}$ and $p_{\mathrm{mic},jt} \times \mathrm{miceffect}_t$ to de-trend these numbers towards an RDT-standardised 2019 benchmark. To reduce the impact of any unmodelled factors contributing short-term temporal fluctuations to the case reports, we then wrap our 4 years of imputed and de-trended data around the calendar year to construct (from each posterior draw) a single year of representative data from the median in each month.

$$\mathrm{cases}_{jt\,:\,\mathrm{where\ case\ data\ non-missing}} \sim \mathrm{NegBin}\left(\begin{array}{c}\mathrm{mean}=I_{jt} \times \mathrm{approx\ catchment\ pop}_{jt}, \\ \mathrm{over\ dispersion\ factor}=\sigma\end{array}\right)$$

$$\log I_{jt} = c + f_{intercept}(\mathrm{loc}_j) + f_{(1)}(\mathrm{loc}_j) \times \beta_{\mathrm{spline},t}^{(1)} + f_{(2)}(\mathrm{loc}_j) \times \beta_{\mathrm{spline},t}^{(2)} + f_{(3)}(\mathrm{loc}_j) \times \beta_{\mathrm{spline},t}^{(3)}$$
$$+ f_{(4)}(\mathrm{loc}_j) \times \beta_{\mathrm{spline},t}^{(4)} + f_{(5)}(\mathrm{loc}_j) \times \beta_{\mathrm{spline},t}^{(5)} + f_s easonal(\mathrm{loc}_j, \mathrm{mod}(t,12)$$
$$+ p_{\mathrm{mic,jt}} \times \mathrm{mic\ effect}_t$$

$$f_{\mathrm{intercept}}(\cdot) \sim \mathrm{Gaussian\ Process}(\mathrm{range}_{\mathrm{intercept}}, \mathrm{scale}_{\mathrm{intercept}})$$

$$f_{(1)}(\cdot), f_{(2)}(\cdot), f_{(3)}(\cdot), f_{(4)}(\cdot), f_{(5)}(\cdot) \sim \mathrm{Gaussian\ Process}(\mathrm{range}_{\mathrm{spline}}, \mathrm{scale}_{\mathrm{spline}})$$

$$f_{\mathrm{seasonal}}(\cdot) \sim \mathrm{Gaussian Process}(\mathrm{range}_{\mathrm{seasonal\ time}}, \mathrm{scale}_{\mathrm{seasonal\ time}}) \otimes \mathrm{Gaussian\ Process}(\mathrm{range}_{\mathrm{seasonal}}, \mathrm{scale}_{\mathrm{seasonal}})$$

$$\mathrm{mic\ effect}_t = \mu + f_{\mathrm{mic}}(t), f_{\mathrm{mic}}(\cdot) \sim \mathrm{AR}_1(\mathrm{scale}_{\mathrm{mic}}, \mathrm{AR\ par}_{\mathrm{mic}})$$

$$\log \mathrm{range}_{\mathrm{intercept}} \sim \mathrm{Normal}(-1, 1^2),\ \log\mathrm{range}_{\mathrm{spline}},\ \log\mathrm{range}_{\mathrm{seasonal}},\ \log\mathrm{range}_{\mathrm{seasonaltime}} \sim \mathrm{Normal}(1, 1^2)$$

$$\log \mathrm{scale}_{\mathrm{intercept}} \sim \mathrm{Normal}(2, 1^2),\ \log\mathrm{scale}_{\mathrm{spline}},\ \log\mathrm{scale}_{\mathrm{seasonal}},\ \log\mathrm{scale}_{\mathrm{seasonaltime}} \sim \mathrm{Normal}(-1, 1^2)$$

$$\log \mathrm{scale}_{\mathrm{mic}} \sim \mathrm{Normal}(-1, 1^2),\ \log\mathrm{it\ AR\ par}_{\mathrm{mic}} \sim \mathrm{Normal}(2, 1^2)$$

$$\log \sigma \sim \mathrm{Normal}(-1, 1^2),\ c \sim \mathrm{Improper\ Uniform}$$

## Model 2
Point-level geostatistical model for approximate case incidence rate at each health facility location used for de-trending (and imputing) the raw monthly case counts towards the production of an RDT-standardised 2019 benchmark.

## Fine-scale prediction of annual incidence surface
The second stage of our inference procedure is to fit a pixel-level geostatistical model with full catchment sub-model to the annual totals at facility level in (each modular posterior draw of) the 12 months of representative counts. On removing the temporal dimension from consideration, it

becomes computationally feasible to allow flexible health facility attractiveness weights in the catchment sub-model and to perform the aggregation of the latent cases from pixel level to facility via this sub-model self-consistently during fitting. In this sense the adopted model structure is at least one step more ambitious than other comparable, multi-scale geospatial models for fine-scale disease mapping from areal-averaged data (*Wilson and Wakefield, 2020*; *Taylor et al., 2018*). Another extension is that we have adopted a spatially varying coefficient (slope) model (*Gelfand et al., 2003*) to describe the relationship between our static, environmental covariates and the log incidence rate. The motivation for this is to limit our exposure to bias in this implicit ecological regression (*Wakefield and Smith, 2016*) due to unmodelled factors, such as the potential role of human movement between regions and spatial variations in the dominant anopheline species. Both of these could lead to differences in the relationship between environmental variables and the case incidence rate amongst the human populations resident in different areas of the country. A decision was made not to attempt to learn a shrinkage hyper-parameter acting on the static covariate slopes in order to avoid exposure to over-shrinkage given that the aggregate dataset may be thought of as inherently under-powered for learning slopes relative to a comparable point-level dataset of similar design and size. Previous applications of fine-scale modelling to aggregate malaria datasets (*Sturrock et al., 2014*; *Alegana et al., 2016*) used aggressive covariate selection approaches, which retained far fewer environmental variables than are typically found to be important for prediction at this scale based on point prevalence surveys (*Bhatt et al., 2015*; *Weiss et al., 2015*).

A lack of data on treatment seeking behaviours for malaria patients in Haiti has previously been identified as a core knowledge gap (*Keating et al., 2008*). As our primary interest here concerns the recovery of accurate spatial patterns of malaria incidence, we are less worried about the overall rate of treatment seeking (which studies in African settings suggest is rarely below 30% for acute febrile illness [*Alegana et al., 2017b*]) than in the possibility of spatial variation. Studies of treatment seeking behaviour in both low- and high-resource settings indicate a tendency for treatment seeking rates to decline with increasing travel-time distance from the nearest point of care (*Alegana et al., 2017b*; *Ensor and Cooper, 2004*). However, very little decline is seen until beyond 100 min travel time in well-studied settings (such as Namibia [*Alegana et al., 2012*]), and at the $1 \times 1$ km resolution of our map making almost 96.4% of pixels with non-zero population density lies within this distance from their nearest health facility. For this reason, we do not anticipate a strong spatial variation in treatment seeking rates across the country due to this effect, but we have nevertheless constructed an access distance-dependent treatment seeking probability map (following the Namibian example, with maximum treatment seeking probability of 65%) as a first-order approximation.

The complete Bayesian model used in this stage is described in hierarchical notation in the box for Model 3 below. Once again a combination of the *TMB* and *INLA* packages are used to fit this model with a Laplace approximation over the random field and the (logarithm of) catchment attractiveness weights, with a Multivariate Normal approximation in the remaining hyper-parameters centred on the empirical Bayes estimator.

$$\text{annual representative cases}_j \sim \text{NegBin}\left( \begin{array}{l} \text{mean} = \text{expected cases}_j, \\ \text{over dispersion factor} = \sigma \end{array} \right)$$

$$\text{expected cases}_j = \sum_i C_{i \to j} \times \text{population}_i \times I_i \times \text{treatment seeing prob}_i$$

$$C_{i \to j} \propto \frac{W_j}{T_{i \to j}^2}, \qquad \log W_j \sim \text{Normal}(0, 0.5^2)$$

$$\log I_i = c + X'_{\text{static}}(\beta_{\text{static}} + f_{\text{static}}(\text{loc}_i)) + f_{\text{intercept}}(\text{loc}_i)$$

$$f_{\text{intercept}}(\cdot) \sim \text{Gaussian Process}(\text{range}_{\text{int}}, \text{scale}_{\text{int}})$$

$$\beta_{\text{static},k} \sim \text{Normal}(0, 1^2), f_{\text{static},k}(\cdot) \sim \text{Gaussian Process}(\text{range}_{\text{covs}}, \text{scale}_{\text{covs}})$$

$$\log \text{scale}_{\text{static}}, \log \text{scale}_{\text{covs}} \sim \text{Normal}(-1, 1^2), \qquad \log \text{range}_{\text{static}}, \log \text{range}_{\text{covs}} \sim \text{Normal}(1, 1^2)$$

$$\log \sigma \sim \text{Normal}(-1, 1^2), \qquad c \sim \text{Improper Uniform}$$

## Model 3

Catchment-based geostatistical model for annual case count at each health facility location used to produce our baseline clinical incidence rate surface.

### Spatio-temporal modelling of seasonal fluctuations in case incidence

In the third and final stage of our inference procedure, we hold fixed the health facility attractiveness weights, baseline incidence surface, and annual (i.e., spatial) over-dispersion factors learnt in the previous step. This allows (at the limit of our computational resources; 128 GB RAM) to model the seasonal variations in incidence at fine-scale in a spatio-temporal geostatistical regression against the monthly case counts in (each modular posterior draw of) the representative dataset. The model structure for the seasonality term is the same as that used in the first stage: a separable (Kronecker product) spatio-temporal Gaussian process with circularity (over the calendar months) in the temporal dimension (an exponential kernel on the circle). Due to computational limitations, a spatially varying slope model was infeasible for the dynamic covariates, hence an ordinary linear regression structure was used instead; again with a fixed, limited amount of prior shrinkage in deference to the limited power provided by the aggregate data. Posterior sampling was conducted exactly as described for stages one and two above with implementation in *TMB* and *INLA*. The full Bayesian hierarchy is described in the box for Model 4 below.

$$\text{monthly reduced cases}_{jt} \sim \text{NegBin}\left(\begin{array}{l} \text{mean} = \text{expected cases}_{jt}, \\ \text{over dispersion factor} = \sigma \end{array}\right)$$

$$\text{expected cases}_{jt} = \sum_i C_{i \to j} \times \text{population}_i \times I_{it} \times \text{treatment seeking prob.}_i$$

$$C_{(i \to j)} \propto \frac{W_j}{T_{i \to j}^2}, \log W_j, \qquad \log I_{\text{baseline}} = \text{fixed from earlier fit}$$

$$\log I_{it} = c + \log I_{\text{baseline}} + X'_{\text{temporal}}\beta_{temporal} + f_{\text{seasonal}}(\text{loc}_i, t)$$

$$f_{\text{seasonal}}(\cdot) \sim \text{Gaussian Process}(\text{range}_{\text{seasonal time}}, \text{scale}_{\text{seasonaltime}}) \\ \otimes \text{Gaussian Process}(\text{range}_{\text{seasonal}}, \text{scale}_{\text{seasonal}})$$

$$\beta_{\text{temporal}} \sim \text{Normal}(0, 1^2), \qquad \log \sigma \sim \text{Normal}(-1, 1^2)$$

$$\log \text{scale}_{\text{seasonal}}, \log \text{scale}_{\text{seasonal time}} \sim \text{Normal}(-1, 1^2), \qquad \log \text{range}_{\text{seasonal}}, \log \text{range}_{\text{seasonal time}} \sim \text{Normal}(1, 1^2)$$

$$c \sim \text{Improper Uniform}$$

## Model 4

Catchment-based geostatistical model for representative monthly case count at each health facility location used to produce our seasonality profile.

The R and TMB codes used for running this analysis are provided for reference as Supplementary Information.

## Acknowledgements

Carl Fayette and Franck Monestime with IMA World Health for assistance with implementation of field surveys. Katherine Pendleton, Blaise Tschirhart, Divya Sukumar, Ashraf Patel, Namratha Kolur, and Camelia Herman for assistance in laboratory serology data collection.

## Additional information

### Funding

| Funder | Grant reference number | Author |
| --- | --- | --- |
| Bill and Melinda Gates Foundation | OPP1152978 | Ewan Cameron<br>Katherine A Twohig<br>Punam Amratia<br>Daniel J Weiss<br>Peter W Gething |

Katherine E Battle

The funders had no role in study design, data collection and interpretation, or the decision to submit the work for publication.

## Author contributions

Ewan Cameron, Data curation, Formal analysis, Supervision, Visualization, Methodology, Writing - original draft, Writing - review and editing; Alyssa J Young, Eric Rogier, Data curation, Validation, Investigation, Project administration, Writing - review and editing; Katherine A Twohig, Data curation, Validation, Project administration; Emilie Pothin, Data curation, Investigation, Methodology; Darlene Bhavnani, Data curation, Investigation, Project administration, Writing - review and editing; Amber Dismer, Data curation, Validation, Investigation, Methodology, Writing - review and editing; Jean Baptiste Merilien, Data curation, Validation, Investigation, Project administration; Karen Hamre, Validation, Investigation, Writing - review and editing; Phoebe Meyer, Data curation, Validation, Investigation, Writing - review and editing; Arnaud Le Menach, Data curation, Supervision, Validation, Investigation, Project administration, Writing - review and editing; Justin M Cohen, Supervision, Funding acquisition, Project administration, Writing - review and editing; Samson Marseille, Data curation, Validation, Writing - review and editing; Jean Frantz Lemoine, Data curation, Funding acquisition, Validation, Investigation, Project administration, Writing - review and editing; Marc-Aurèle Telfort, Data curation, Validation, Project administration, Writing - review and editing; Michelle A Chang, Funding acquisition, Validation, Project administration, Writing - review and editing; Kimberly Won, Data curation, Writing - review and editing; Alaine Knipes, Writing - review and editing; Punam Amratia, Formal analysis, Validation, Investigation, Methodology, Writing - review and editing; Daniel J Weiss, Resources, Data curation, Writing - review and editing; Peter W Gething, Resources, Supervision, Funding acquisition, Methodology, Writing - review and editing; Katherine E Battle, Data curation, Formal analysis, Validation, Methodology, Project administration, Writing - review and editing

## Author ORCIDs

Ewan Cameron (iD) https://orcid.org/0000-0002-8842-3811

## Decision letter and Author response

Decision letter https://doi.org/10.7554/eLife.62122.sa1
Author response https://doi.org/10.7554/eLife.62122.sa2

# Additional files

## Supplementary files
- Source code 1. R script for Stepwise Modelling of Case Count Dataset.
- Source code 2. TMB code for the Impact of Diagnostic Type.
- Source code 3. TMB code for Detrending Model.
- Source code 4. TMB Code for Seasonality Model.
- Source code 5. TMB Code for Static Risk Surface Model.
- Transparent reporting form

## Data availability

The routine case data at health facility level are the property of the Haitian Programme National de Contrôle de la Malaria and are not to be made publicly available at this level of resolution to respect privacy. However, the administrative level summaries are published through the World Malaria Report each year. All covariates used are publicly available through their respective online portals. The TAS serology data used in validation are available upon request as per https://www.nature.com/articles/s41598-020-65419-w#data-availability.

The following datasets were generated:

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
