## [Decision Letter]

**Acceptance summary:**

The manuscript presents detailed findings from a multi-stage Bayesian approach to estimate spatial and spatio-temporal trends in malaria incidence in Haiti. The proposed methodology represents an advance over their well-established methodology, by explicitly modeling catchment areas, that can be adapted for different diseases to inform decision making.

**Decision letter after peer review:**

Thank you for submitting your article "Mapping the endemicity and seasonality of clinical malaria for intervention targeting in Haiti using routine case data" for consideration by *eLife*. Your article has been reviewed by 2 peer reviewers, and the evaluation has been overseen by a Reviewing Editor and a Senior Editor. The following individual involved in review of your submission has agreed to reveal their identity: Nicole White (Reviewer #1).

The reviewers have discussed the reviews with one another and the Reviewing Editor has drafted this decision to help you prepare a revised submission.

Summary:

The manuscript presents detailed findings from a multi-stage Bayesian approach to estimate spatial and spatio-temporal trends in malaria incidence in Haiti. The proposed methodology represents an advance over their well-established methodology, by explicitly modeling catchment areas, that can be adapted for different diseases to inform decision making. The authors should be commended for the breadth of work undertaken, and methods are justified well overall. However, we have several concerns about how the methods and results are presented and it is not entirely clear how the maps produced by this new model are an improvement over previous simpler models, and of the value this model adds beyond can be inferred directly from case data.

Essential revisions:

1. Even though the manuscript presents methodological advances in risk mapping, it lacks demonstration of the value added by the new models over simpler risk maps or over incidence patterns alone. The maps in Figure 1 seem to recapitulate the patterns in the case data. This suggests a good fit of the model, but it does not address the question of what value this model adds beyond what could be gleaned from examining the case data alone. Similarly, it is not clear whether Figure 3 tells us anything different than we might already know by mapping incidence at those administrative levels. It would be useful to provide concrete examples of the value added (beyond the discussion in lines 262-272) by these models. How simpler methodologies may be misleading with respect to transmission risk, or how these new maps lead to different decisions as compared to simpler approaches.

2. Related to the above – In terms of the methodological advance about incorporating catchment areas, it would be more convincing performance was compared to an alternative model without that feature. Likewise for the part about seasonality or any other advances presented. Demonstrating improved performance (in some way or another) is a routine expectation when introducing a newly "improved" predictive model.

3. Related to the above, it could be argued that the value added by these maps relies on their granularity. However, the validity of estimates is not validated at this fine scale. This should be explicitly discussed.

4. The paper is well written, but the significance of the advances made here is not made as clear as I think it should be for a general audience. Someone not familiar with malaria in Haiti or geospatial statistics should have an easier time understanding why this work is significant.

5. Given the complexity of the proposed methodology the presentation of the various modelling stages is difficult to follow in the Materials and methods section. This can be improved (e.g. no explanation of different subscripts; common parameters shared by the different models).

6. Please make code (and data) available. Per *eLifes* policy, all data needed to reproduce the study findings should be made available upon publication

7. The presentation of findings from the spatially-varying regression focus on the covariate(s) with the dominant positive and negative effect (Figure 4 and Figure 7). How was dominance defined, given that different covariates appear to be a mix of continuous and binary variable defined on different scales (or were variables standardised as part of processing)? Similarly, was posterior uncertainty in parameter slopes accounted for when determining dominance? I understand that these results are not intended to have a causal interpretation, however further details about chosen metrics here would help with interpretation.

8. The discussion of results around health facility catchments (p9, start line 168) bears resemblance to gravity modelling, yet the manuscript does not appear to reference related research. Given the proposed catchment model is a core element of the proposed methodology, further explanation of how this applies and/or builds upon existing gravity-based approaches would help place the manuscript in the context of related research.

9. The Materials and methods present four stages of modelling that are inter-related (e.g. imputed values of p_{mic,jt} from Model 1 are treated as a covariate in the geostatistical model defined in Model 2). The presentation of the full approach is quite dense and I found myself constantly switching between Models to identify connections/shared parameters. For example, I was unsure how are Models 2 and 3 are connected, or are they separate models? To improve the clarity of presentation, an overarching figure denoting links between modelling stages would be useful.

10. The introduction cites a lack of information to estimated treatment seeking propensity as a challenge for fine-scale disease mapping. This is again discussed in the Materials and methods section (lines 493 to 505), however I did not understand how this was accounted for in Model 3; were these probabilities/propensities treated as known quantities in Model 3 or were they estimated (with uncertainty)? Some further details are provided in the Discussion about this quantity (lines 281 to 284), however specific details were lacking in the presentation of Model 3 on p 25 (line 511).

11. 122: The first two paragraphs of the results were purely descriptive and read like figure captions.

12. 80-81: How accurate and durable are these serological assays following infection?

13. 421: It is not clear how necessary the seasonal trend in microscopy vs RDT is to model and how reliable it is. There appears to be one year (2019) in which there is a comparison to an RDT "gold standard" (not sure how this comparison works), but the premise of why that should be representative of other years and why there should be seasonality in this in the first place is unclear.

14. 480-484: I can understand how human movement and vector species variations could result in problems with the model, but I am less convinced that allowing slopes on linear relationships of environmental variables is an adequate way to address this problem.

15. 102: "clean the data of epidemic fluctuations" is an odd way to put it. If a temporally aggregated or de-trended description of incidence is desired, that's fine. But epidemic fluctuations are in fact real and therefore not something to be cleaned.

16. Paragraph beginning on 122: While it is good that the predicted case patterns at fine-scale resembled the patterns of cases across the country, it is not clear what new information is gained by the very sophisticated mapping exercise in this manuscript. It would seem that this paragraph could be written based on case data alone.

17. 149: While I agree with everything written in this paragraph, the results it presents are underwhelming. Given that the relative importance of different variables is only interpretable within the context of this model, what value is there in these results beyond the limited context of this model?

18. Figure 6: In addition to this figure, it might be helpful to have one that shows these patterns over time somehow (i.e., time on the x-axis).

[Editors' note: further revisions were suggested prior to acceptance, as described below.]

Thank you for resubmitting your work entitled "Mapping the endemicity and seasonality of clinical malaria for intervention targeting in Haiti using routine case data" for further consideration by *eLife*. Your revised article has been evaluated by a Reviewing Editor and a Senior Editor.

We think the additional analyses presented (Figure 9 and associated text) are a good addition to the manuscript. However, before accepting the manuscript, we ask you to please incorporate them into the appropriate sections in the manuscript (these are mostly results not methods).

---

## [Author Response]

Essential revisions:1. Even though the manuscript presents methodological advances in risk mapping, it lacks demonstration of the value added by the new models over simpler risk maps or over incidence patterns alone. The maps in Figure 1 seem to recapitulate the patterns in the case data. This suggests a good fit of the model, but it does not address the question of what value this model adds beyond what could be gleaned from examining the case data alone. Similarly, it is not clear whether Figure 3 tells us anything different than we might already know by mapping incidence at those administrative levels. It would be useful to provide concrete examples of the value added (beyond the discussion in lines 262-272) by these models. How simpler methodologies may be misleading with respect to transmission risk, or how these new maps lead to different decisions as compared to simpler approaches.2. Related to the above – In terms of the methodological advance about incorporating catchment areas, it would be more convincing performance was compared to an alternative model without that feature. Likewise for the part about seasonality or any other advances presented. Demonstrating improved performance (in some way or another) is a routine expectation when introducing a newly "improved" predictive model.

The accuracy of diagnostic services, access to treatment for malaria patients, and the fidelity of data reporting from healthcare providers in Haiti have not always been adequate for epidemiological monitoring. There are a number of known issues in particular that have forewarned malariologists and policy makers from attempting to produce operational risk stratifications according to “what could be gleaned from examining the case data alone” or “by mapping incidence at those administrative levels”.

A 1998 study by Kachur et al., (Pan Am J Public Health) compared the diagnostic skills of in-country microscopists against those of world-experts from the US Centres for Disease Control: of the 207 smears identified as *Pf* positive by the former, only 46 were confirmed positive by the latter, implying a predictive value positive of just 22.2%. A high false positive rate under in-country microscopy was also proposed by Weppelmann et al., (2018; PLoS ONE) as a likely explanation for the decline in recorded malaria incidence by 40% in Haiti over the period 2010 to 2015 during which natural disasters caused widespread disruption and multiple disease outbreaks; the post-earthquake period being characterised by a necessitated transition from in-country microscopy to standardised RDT diagnosis at many local healthcare facilities. In 2019 the overall proportion of positive malaria diagnoses made by in-country microscopists was 15%, with substantial regional variations around this mean; modelling the plausible rate of false positive diagnoses among this fraction of case reports was thus considered an indispensable step towards estimating the transmission pattern of malaria in Haiti.

Although substantial efforts have been made to improve record keeping and the reporting of data from local healthcare providers, there remains a non-negligible degree of missingness (22.5%) in the case report dataset. Given a noisy response and a seasonal transmission context it would be difficult to justify any simple approach to data imputation, such as the fitting of per-facility constant, linear, or spline regression models to non-missing months. Similarly, the interpretation of incidence patterns without some mechanism to account for treatment seeking behaviours (namely, a catchment model) has also been explicitly cautioned against by Boncy et al., (2015, Malaria Journal), who note: “A critique of current reporting methods is that cases are reported based on the hospital where the diagnosis occurred—not where the patient actually resides (which is often very far away from the hospital). This biases the statistics which consequently impacts where malaria elimination strategies are targeted.”

A note regarding existing risk maps. Relative to most other malaria endemic countries, Haiti represents a low transmission setting, with one consequence being that it is prohibitively expensive to power a cross-sectional parasite prevalence survey to resolve spatial patterns of malaria prevalence. For instance, there were only 12 *Pf* positive individuals identified by RDT out of the 14,795 children tested in the Transmission Assessment Survey (TAS; 2012 and 2014; Knipes et al., 2017, PLoS NTDs). As a result, there is a paucity of local, survey-based malariametric datasets against which a conventional geospatial model could be fit to infer the spatial pattern of risk. Leveraging correlations between environmental covariates and malaria risk in nearby countries, Weiss et al., (2019; Lancet) offer a prevalence-based risk map for Haiti (with large uncertainties) for which the median surface post-2012 fails to identify either the Grand’Anse or the Central Valley as hotspots of transmission (in direct contradiction to outputs from the present study, to local knowledge, and the TAS survey seropositivity results).

A note on Figure 1. When comparing the fine scale map against the health facility data on this Figure, it is worth bearing in mind that the representative case totals shown here have already been through two steps of modelling (detrending and filtering, and micro-to-RDT standardisation).

For all the reasons given above we do not consider our method to be “improved” over a previously existing method; rather we see it as at last providing a solution to a problem that was hitherto unsolved. To address the reviewers’ comments in this light we have amended the main text to better contextualise the ignorance of transmission patterns in Haiti prior to this study and to explain the necessity of a complex model-based approach to parse the available data.

New paragraph added to the Introduction (page 4): “Setting aside the challenge of catchment modelling for fine scale risk mapping, in the case of malaria transmission in Haiti even a coarse quantification of the sub-national patterns and trends in risk has hitherto proven elusive owing to additional limitations of the case report dataset. Foremost of these has been concerns over variability in the specificity of the applied diagnostic methods. A 1998 study by Kachur et al. [30] compared the classifications of local microscopists against those of world-experts from the US Centres for Disease Control: of the 207 blood smears identified as parasite positive by the former, only 46 we confirmed positive by the latter, corresponding to a positive predictive value of just 22.2%. A systematic transition from microscopy to standardised RDT diagnosis at many Haitian healthcare providers has been identified as a likely explanation for the recorded 40% decline in the national malaria case count over the period 2010 to 2015 during which natural disasters disrupted critical supply chains and triggered multiple disease outbreaks [31]. Microscopic diagnoses contributed 15% of reported malaria cases nationally in 2019, with substantial regional variation around this mean. Another limitation that has obstructed risk mapping from the case report dataset is that, despite recent efforts to improve the national health surveillance system, there remains substantial missingness (22.5%) in the record of case totals by facility.”

3. Related to the above, it could be argued that the value added by these maps relies on their granularity. However, the validity of estimates is not validated at this fine scale. This should be explicitly discussed.

In the original manuscript we discussed the challenges of validation for ‘disaggregation models’ (such as that of the present study) for which there is no direct validation data available: (“Validation of model outputs from this class of ‘down-scaling’ models is also uniquely challenging; the hold-out of aggregate response data is of limited utility for testing fine-scale accuracy, since only ancillary pointlevel observations can overcome the potential for ‘ecological fallacy’ [26].”)—and we described a validation exercise using the sero-prevalence measurements from the TAS dataset (sub-section entitled “Validation against a serology dataset”). As was noted therein the sample drawn from each school was restricted to children attending on the survey day aged 6-7 y/o; however, we did not explicitly highlight that an underlying assumption of this validation exercise was that the school catchments represent aggregations over geographic areas substantially smaller than those of the health facility catchments, albeit not being strictly a point-level dataset. We now explicitly quantify the relative difference in areal scale between the health facility catchments for malaria.

The Easy Access Group (EAG) survey has studied malaria prevalence and sero-prevalence among school children, health facility attendees, and church attendees in the Grand’Anse and Artibonite (Druetz et al., 2020, BMC Medicine). In each group, all RDT positive individuals and 20% of RDTnegative enrolled individuals were followed to their home location by a survey team member and a precise location was recorded via GPS. Using this dataset (priv. comm.) one can estimate the relative catchment sizes of all-age, malaria-positive febrile individuals seeking care at local health facilities (as per the routine case data) and 6-7 y/o children attending a local school (as per the TAS dataset), respectively. The median Euclidean distance in each case is 0.79 km and 1.75 km, implying that the areal aggregation function of the school-based sample is approximately a factor of 5 smaller than that of the health facility dataset. Worth noting is that in both cohorts there is a long tail in the distribution of venue-to-household distances.

These points are now noted in the revised manuscript in a new paragraph on page 16: “Importantly, although the TAS sample cohort of primary school children is not strictly a point-level dataset, it is expected that on average the school catchments represent a much smaller areal aggregation than the health facility catchments of the case report dataset, and as such provide a degree of validation of the sub-catchment resolution accuracy of the down-scaling model. To make precise this difference we compare the median household-to-school distance (Euclidean) of children in the Easy Access Group (EAG) study of Druetz et al. [37] against those of parasite positive, febrile individuals presenting to health facilities in the same dataset (priv. comm.). The results of 0.79 km and 1.75 km, respectively, suggest that (at least for the Grand’Anse and Artibonite focus areas of the EAG study) the areal scale of the health facility catchments is larger by a factor of 5.”

4. The paper is well written, but the significance of the advances made here is not made as clear as I think it should be for a general audience. Someone not familiar with malaria in Haiti or geospatial statistics should have an easier time understanding why this work is significant.

In responding to comments 1 and 2 above we have added substantial contextual information highlighting the advances made in this paper in terms of providing, for the first time, an actionable statistical representation of the spatial endemicity and seasonality of malaria transmission in Haiti. We have also added the following in the Discussion (page 17) to highlight the improvement over the non-case report based map of Weiss et al., (2019): “Another point of comparison that demonstrates the value of this work is provided by the malaria risk map for Haiti produced by Weiss et al. [11] within the scope of a global malaria mapping initiative. Owing to the paucity of survey-based prevalence data for Haiti this traditional geospatial model must rely on correlations between environmental covariates and malaria risk seen in other countries to produce an estimate. The resulting median surface for 2017 fails to identify either the Grand’Anse or the Central Valley as hotspots of transmission, in direct contradiction to outputs from the present study, local knowledge, and the TAS sero-positivity results.”

5. Given the complexity of the proposed methodology the presentation of the various modelling stages is difficult to follow in the Materials and methods section. This can be improved (e.g. no explanation of different subscripts; common parameters shared by the different models).

We have now added explanations of the subscripts (i -> pixels, j -> facilities, t -> month) in the Materials and Methods. Further clarity on the model details is provided in responses to specific comments below (namely: 9 and 13).

6. Please make code (and data) available. Per eLifes policy, all data needed to reproduce the study findings should be made available upon publication

This will be done.

7. The presentation of findings from the spatially-varying regression focus on the covariate(s) with the dominant positive and negative effect (Figure 4 and Figure 7). How was dominance defined, given that different covariates appear to be a mix of continuous and binary variable defined on different scales (or were variables standardised as part of processing)? Similarly, was posterior uncertainty in parameter slopes accounted for when determining dominance? I understand that these results are not intended to have a causal interpretation, however further details about chosen metrics here would help with interpretation.

Variables were standardised prior to fitting (this is now noted explicitly under Materials and Methods: “All covariates were standardised towards a standard Normal distribution by subtracting the empirical mean and dividing by the empirical standard deviation (following a logarithmic transformation for long-tailed, positive covariates, such as elevation).”), but this is not necessary knowledge for interpretation of Figures 4 and 7 in which the dominant covariate is defined as the covariate with greatest influence on the linear predictor in each pixel. That is, covariate times slope, rather than just slope (as the reviewer might have understood it to be). To avoid this confusion notes are added to this effect in the Results section, along with a note to confirm that this is indeed a representation from the full posterior (most frequently dominant covariate over posterior draws): “Here ‘dominant’ is to be understood as having the greatest contribution to the linear predictor of the model (i.e., slope × covariate value; not just slope); and results are presented as a median summary of the dominant covariate in each pixel over all posterior draws.”

8. The discussion of results around health facility catchments (p9, start line 168) bears resemblance to gravity modelling, yet the manuscript does not appear to reference related research. Given the proposed catchment model is a core element of the proposed methodology, further explanation of how this applies and/or builds upon existing gravity-based approaches would help place the manuscript in the context of related research.

The reviewer is correct that our travel time model bears a similarity to gravity modelling; in the original manuscript we noted this explicitly (calling it “a gravity-style representation”) and referred to the “alternatives of Euclidean (‘as-the-crow-flies’) distances and/or hard (tessellation-style) boundaries”. We also cited a number of works in this area (Duncan et al.,: thought see note on Minor Revisions 19; and Nelli et al.,).

9. The Materials and methods present four stages of modelling that are inter-related (e.g. imputed values of p_{mic,jt} from Model 1 are treated as a covariate in the geostatistical model defined in Model 2). The presentation of the full approach is quite dense and I found myself constantly switching between Models to identify connections/shared parameters. For example, I was unsure how are Models 2 and 3 are connected, or are they separate models? To improve the clarity of presentation, an overarching figure denoting links between modelling stages would be useful.

The overview modelling strategy is presented under “Stepwise modelling approach designed for robustness against unmodelled sources of noise” in the Materials and Methods section. As explained there, each stage of the modelling is conducted stepwise in the order specified and uncertainties are propagated forwards as per a ‘cut’ model structure. While a schematic figure could be included (and would be useful for oral presentations) it may appear a little simplistic in the manuscript context as there would be only a set of four boxes with connections from each one to the next.

10. The introduction cites a lack of information to estimated treatment seeking propensity as a challenge for fine-scale disease mapping. This is again discussed in the Materials and methods section (lines 493 to 505), however I did not understand how this was accounted for in Model 3; were these probabilities/propensities treated as known quantities in Model 3 or were they estimated (with uncertainty)? Some further details are provided in the Discussion about this quantity (lines 281 to 284), however specific details were lacking in the presentation of Model 3 on p 25 (line 511).

The treatment seeking surface is constructed following a heuristic formulation designed to capture the general relationship between treatment seeking propensity and travel time to nearest health facility seen in other endemic settings (as described in the Materials and Methods section; formerly lines 493 to 505; as noted by the review). This surface is treated here as a fixed input (meaning no propagation of uncertainties) and the lack of local data to inform this surface is noted as a caveat in the Discussion. We have experimented with a variety of versions of this treatment seeking heuristic but find that the impact on the output maps is modest relative to the other uncertainties. To clarify this point we have added a note in the presentation of Model 3: “This treatment seeking surface is deployed here as a fixed input without any explicit uncertainty propagation; however, we note that plausible variations to the assumed form yield differences to the output incidence rates that are either small or moderate relative to the other uncertainties carried forward and accumulated in this modelling step.”.

11. 122: The first two paragraphs of the results were purely descriptive and read like figure captions.

Here we have followed the style of other manuscripts in *eLife* and similar scientific journals, in which the presentation in the Results section is rather ‘dry’ while more ‘colourful’ phrasing is reserved for the Discussion.

12. 80-81: How accurate and durable are these serological assays following infection?

On an individual level these serological assays have a poor diagnostic (sensitivity/specificity) performance (as we have also seen this year in the case of SARS-CoV-2!), but have a well-established role for assessing spatio-temporal variations in malaria transmission intensity at the community level: see, e.g., Sturrock et al., (2016, Trends in Parasitology); Surendra et al., (2020, BMC Medicine); Ashton et al., (2015, Am J Trop Med Hyg).

13. 421: It is not clear how necessary the seasonal trend in microscopy vs RDT is to model and how reliable it is. There appears to be one year (2019) in which there is a comparison to an RDT "gold standard" (not sure how this comparison works), but the premise of why that should be representative of other years and why there should be seasonality in this in the first place is unclear.

There is no explicit seasonal trend in microscopy vs RDT in the model, rather the effect is represented as the realisation of an AR1 process with monthly timesteps and empirical Bayes hyperparameter learning. There is no “gold standard” year, rather the effect is learned as any other regression slope within a Bayesian model setting: where there is a large (small) association of higher case reports per unit population with higher use of microscopy the learned slope will be large (small). The allowance for a smooth variation over time is to acknowledge the diagnostic specificity may be changing over time as the pool of technicians changes composition or skill.

14. 480-484: I can understand how human movement and vector species variations could result in problems with the model, but I am less convinced that allowing slopes on linear relationships of environmental variables is an adequate way to address this problem.

The value of mixed effects models to assist with ecological regressions when the covariate set fails to include key drivers of spatial heterogeneity has been explored by Hamil et al., (2015, Landscape Ecology). We now add a reference to this work at this location.

15. 102: "clean the data of epidemic fluctuations" is an odd way to put it. If a temporally aggregated or de-trended description of incidence is desired, that's fine. But epidemic fluctuations are in fact real and therefore not something to be cleaned.

We have modified this to “dampen epidemic fluctuations relative to long-term transmission potential”.

16. Paragraph beginning on 122: While it is good that the predicted case patterns at fine-scale resembled the patterns of cases across the country, it is not clear what new information is gained by the very sophisticated mapping exercise in this manuscript. It would seem that this paragraph could be written based on case data alone.

See the response to comments 1 and 2.

17. 149: While I agree with everything written in this paragraph, the results it presents are underwhelming. Given that the relative importance of different variables is only interpretable within the context of this model, what value is there in these results beyond the limited context of this model?

We do not claim value beyond the context of this model, but hope that its value within the context itself (i.e., helping readers to understand which covariates are driving the fine scale patterns locally) is sufficient to warrant its inclusion in the manuscript.

18. Figure 6: In addition to this figure, it might be helpful to have one that shows these patterns over time somehow (i.e., time on the x-axis).

Since one cannot practically plot a time series for every pixel the reviewer’s suggestion would require an aggregation to suitable administrative scale with some decisions around population weighting and transformation (geometric or arithmetic mean of incidence effectively). The particularly interesting feature of the seasonality reconstructions here are their heterogeneity, so we prefer not to confuse the picture with areal aggregations of this kind.

[Editors' note: further revisions were suggested prior to acceptance, as described below.]

We think the additional analyses presented (Figure 9 and associated text) are a good addition to the manuscript. However, before accepting the manuscript, we ask you to please incorporate them into the appropriate sections in the manuscript (these are mostly results not methods).

We have moved the comparison against naive catchment models from the Methods and Materials section to the Results section, and minor adjustments accordingly.